# OPENDATABENCH: REAL-WORLD BENCHMARK FOR TABLE INSIGHT GENERATION AND QUESTION ANSWERING OVER OPEN DATA

## ABSTRACT

The promise of Large Language Models (LLMs) for data analysis is hindered by benchmarks that inadequately reflect real-world complexities, including multiple large tables and external knowledge. Moreover, they mainly focus on fact retrieval via Question Answering (QA) and overlook the critical task of exploratory insight generation. To address these gaps, we introduce *OpenDataBench*, a benchmark built from governmental open data capturing these practical challenges. It features two types of tasks: multifaceted *Table QA* tasks that require answering complex decomposable questions with either text or graphs, and *Table Insight* tasks that challenge models to generate expert-level findings from exploratory data analysis. We evaluate state-of-the-art LLMs and our proposed agentic solution on OpenDataBench. Our experimental results indicate that even top-performing models struggle with both tasks. This highlights a significant gap between current model capabilities and the demands of realistic data analysis. OpenDataBench serves as a rigorous benchmark for advancing research on LLM-driven data analysis systems capable of addressing both reactive question answering and proactive insight discovery. Code and sample data are available at `https://anonymous.4open.science/r/opendatabench-8AFA/`.

## 1 INTRODUCTION

The ability to reason over structured data is a cornerstone of modern data science and a long-standing challenge in artificial intelligence. With the advent of Large Language Models (LLMs), we have witnessed a paradigm shift in how humans interact with complex information (OpenAI et al., 2024a; Team et al., 2023; Qwen et al., 2025; DeepSeek-AI et al., 2025). These models have led to the development of sophisticated agents designed to democratize data analysis, promising a future where any user can pose natural language questions to a dataset and receive accurate answers (Hu et al., 2024; Su et al., 2024; Wang et al., 2024). The ultimate vision is an autonomous system that not only retrieves information but also uncovers the knowledge hidden within raw data.

However, a significant gap persists between this vision and reality, as the benchmarks lack real-world complexity. While datasets like WikiTableQuestions (Pasupat & Liang, 2015) and Spider (Yu et al., 2018) propelled research in semantic parsing and text-to-SQL, their controlled environments use small-scale tables. They largely neglect practical challenges such as massive table scales, the need to merge multiple tables, and the essential role of metadata and external knowledge. Beyond these data limitations, existing benchmarks have focused primarily on direct fact retrieval (Wu et al., 2025b; Hu et al., 2024). Tasks like QA and text-to-SQL are about retrieving information in response to a query, while missing the capability of proactive insight discovery that data analysts exhibit. Consequently, this discovery-oriented skill remains largely unevaluated, as few benchmarks have formalized insight generation as a primary task (Sahu et al., 2025; Seo et al., 2025).

To bridge these notable gaps in both data realism and task scope, we introduce *OpenDataBench*, a comprehensive benchmark sourced from public repositories like Data.gov. Our benchmark features two complementary tasks: Table Question Answering (*Table QA*) and Table Insight Generation (*Table Insight*), as illustrated in Figure 1. The Table QA task assesses factual reasoning over decomposable questions that explicitly include multiple sub-questions, requiring models to produce

Figure 1: OpenDataBench evaluates LLMs and agents on two table reasoning tasks using large, multi-table datasets supplemented with metadata and external knowledge. (a) The *Table QA* task requires models to answer simple or decomposable questions with textual or visual answers. (b) In contrast, the *Table Insight* task challenges models to perform open-ended exploratory analysis, proactively generating a list of insights and a summary without a specific user query.

answers either in text or in visualizations. In contrast, the Table Insight task challenges models to perform expert-level insight derivation, requiring in-depth analysis and the discovery of trends.

For the Table QA task, we use LLMs, aided by a novel table serialization, to generate a diverse corpus of QA pairs that are then meticulously verified by human annotators. For the Table Insight task, we address the challenge of subjectivity by using the expert-authored reports accompanying datasets as a ground truth. This process yields a benchmark that surpasses existing alternatives by holistically combining the complex Table QA and Table Insight with the diverse data complexities, such as large-scale, multi-tabular datasets that require external knowledge, as detailed in Table 1.

We propose two novel agentic solutions: an *Answer Agent* with fail-safe modules for Table QA, and an *Insight Agent* for Table Insight employing a graph-based exploration process for diverse insight generation. Through comprehensive evaluation, our proposed agents outperform state-of-the-art LLMs and existing agents on both tasks in OpenDataBench. Finally, we provide a detailed qualitative analysis to offer the community clear guidance on key areas for future improvement.

In summary, our paper makes the following four contributions:

• We introduce *OpenDataBench*, featuring tasks for both multifaceted Question Answering and Insight Generation with ground-truths annotated by domain experts. These tasks handle large, multi-tabular, and heterogeneous datasets representing complexity in real-world scenarios.

• We propose two novel agentic solutions: an *Answer Agent* with the task-specific agentic assistance and an *Insight Agent* that uses a graph-based exploration process to generate diverse insights while ensuring correctness by incorporating the Answer Agent.

• Comprehensive evaluation of state-of-the-art models reveals a crucial performance gap, as even top models achieve low accuracy. This underscores the benchmark's difficulty and its alignment with actual challenges.

• We provide a detailed qualitative analysis that categorizes common failure points, offering a clear guide for the community to focus on key areas for model improvement.

## 2 OVERVIEW OF OPENDATABENCH

OpenDataBench is a new benchmark designed to evaluate tabular reasoning in realistic scenarios. The data is derived from governmental open data portals (e.g., Data.gov, Data.gov.uk), which host publicly available datasets from official institutions. Datasets on these portals reflect real-world complexity, consisting of a single or multiple tables containing a large number of records, and rich contextual information. This context is provided through metadata, such as textual descriptions of the dataset, and often supplemented with external knowledge like data dictionaries. The benchmark is designed to evaluate performance on two core data science tasks: Table QA and Table Insight.

**Table QA**: This task requires models and agents to answer simple or decomposable questions with multiple sub-questions. The answers are provided in text or visualizations.

| Benchmark | Insight | QA | | Dataset Characteristics | | | |
|---|---|---|---|---|---|---|---|
| | | Decomposable QA | Visualization | Large-table | Multiple Tables | Metadata | External Knowledge |
| *Existing Benchmarks for Table Question and Answering* | | | | | | | |
| WTQ (Pasupat & Liang, 2015) | ✗ | ✗ | ✗ | ✗ | ✗ | ✗ | ✗ |
| OTT-QA (Chen et al., 2021) | ✗ | ✗ | ✗ | ✗ | ✓ | ✓ | ✓ |
| FeTaQA (Nan et al., 2022) | ✗ | ✗ | ✗ | ✗ | ✗ | ✓ | ✗ |
| DataBench (Osés Grijalba et al., 2024) | ✗ | ✗ | ✗ | ✓ | ✗ | ✗ | ✗ |
| TableBench (Wu et al., 2025b) | ✗ | ✗ | ✓ | ✗ | ✗ | ✗ | ✗ |
| MMQA (Wu et al., 2025a) | ✗ | ✗ | ✗ | ✗ | ✓ | ✗ | ✗ |
| *Existing Benchmarks for Table Insight Generation* | | | | | | | |
| InsightBench (Sahu et al., 2025) | ✓ | ✗ | ✗ | ✗ | ✓ | ✗ | ✗ |
| MT-RAIG (Seo et al., 2025) | ✓ | ✗ | ✗ | ✗ | ✓ | ✓ | ✗ |
| **OpenDataBench** | ✓ | ✓ | ✓ | ✓ | ✓ | ✓ | ✓ |

Table 1: Comparison of existing Table QA and Table Insight benchmarks with respect to task coverage and dataset characteristics.

**Table Insight**: This task challenges models and agents to perform exploratory analysis, generating substantive insights directly from the tabular data without an explicit user query.

### 2.1 BENCHMARK CONSTRUCTION

The construction of OpenDataBench involved a three-stage process as shown in Figure 2: 1) curating datasets from open data portals through a systematic filtering process, 2) annotating QA pairs via a human-in-the-loop, and 3) compiling ground-truth insights by leveraging professional reports.

#### 2.1.1 DATA CURATION

Given the vast and decentralized nature of open data available online, a systematic collection and filtering process was imperative. Our process began by identifying 53 open data platforms with English as the primary language (a complete list is provided in the Table 5). We downloaded all available datasets from these platforms and then applied a series of filtering criteria including at least one of the tabular files covering over 5,000 records and metadata with description to understand the context of the dataset. A more detailed filtering procedure is outlined in the Appendix B.1.2.

#### 2.1.2 ANNOTATION OF QUESTION AND ANSWER PAIRS

To construct a high-quality, complex, and challenging set of QA pairs at scale while mitigating the need for resource-intensive manual annotation, we designed a four-stage generation pipeline that leverages LLMs with human-in-the-loop verification. Inspired by prior work (Wu et al., 2025b), this approach ensures both diversity and correctness. The procedure is detailed below.

1. **Question Generation**: The initial stage focused on generating a diverse pool of candidate questions. To guide the output of the LLM, we first defined eight question types as presented in Appendix B.2.1. These types encompassed not only simple queries (e.g. Ranking, Aggregation) shared with existing benchmarks (Wu et al., 2025b;a) but also complex decomposable questions. The prompt contains the table contents, the title and description of the dataset from the metadata, external knowledge when available, and the designated question type. Generating diverse and meaningful questions with LLMs requires providing them with a representative view of the table contents and value distributions. While prior works mainly focus on smaller datasets such that the entire table or the first several rows could be embedded into the prompt (Wu et al., 2025b), we cannot apply similar approaches to tables with millions of records in our benchmark as it exceeds the limits of LLM context window. To address this challenge, we propose a technique called *Feature type-specific table serialization*, which creates a compact yet informative summary of a table by representing each column according to its data type. For instance, instead of listing all values in a categorical column, we provide only the set of unique categories. This serialization is a core component of our workflow, and a detailed description of the logic for various feature types is presented in Appendix B.2.2. To mitigate model-specific biases in the generated questions, we employed an ensemble of four high-performance LLMs: GPT-4o, GPT-4o-mini (OpenAI et al., 2024b), Gemini 2.0 Flash (DeepMind, 2024), and Gemini 1.5 Pro (Team & et al, 2024).

2. **Question Scoring**: Following generation, the candidate questions underwent an automated scoring and selection phase. Each question was evaluated against four qualitative criteria: relevance to the dataset, its potential to yield insightful or actionable information, sufficient analytical com-

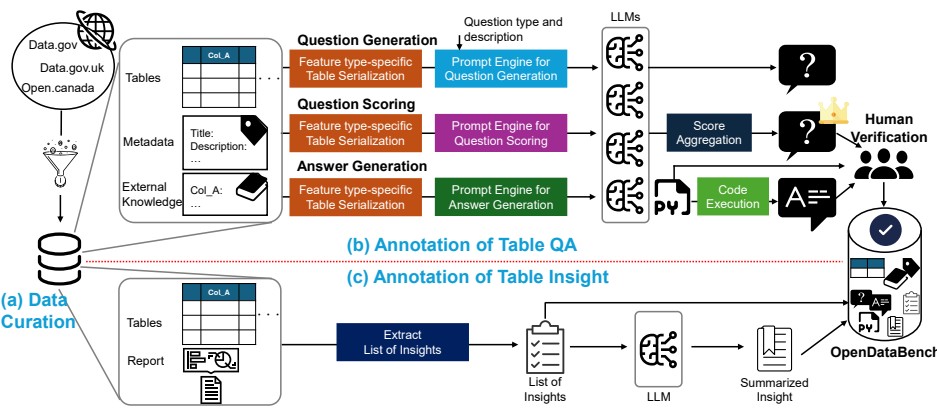

Figure 2: Overview of the three-stage construction process of OpenDataBench

plexity with novelty, and clarity expressed in natural language. In an approach to quality filtering, we tasked each of the four aforementioned LLMs with acting as a judge, selecting its top-5 preferred questions from the generated pool based on the above criteria for each dataset. A score from 5 (highest) to 1 (lowest) was assigned to these selections. The scores from all four models were then aggregated for each question, resulting in a maximum possible score of 20. Based on this aggregated score, we selected the top-10 questions to proceed to the next stage.

3. **Answer Generation**: For each of the top-10 questions per dataset, an LLM was prompted to generate Python code that produces the correct answer. After executing the code from all four models, we measured the answer consensus to filter out questions that yielded unanimous agreement across all four LLMs. Such instances were deemed to indicate a low level of analytical complexity (e.g. single column filtering or aggregation), making them unsuitable for a benchmark in a real-world setting.

4. **Human Verification**: The remaining candidate QA pairs were subjected to a human verification and refinement process. Using a custom-developed annotation GUI as shown in Figure 7, human annotators with expertise in data analysis reviewed each component. Their tasks included: (1) revising the natural language question for clarity and precision; (2) validating, debugging, and refining the Python code for correctness and efficiency; and (3) verifying the final answer derived from the code. During this stage, annotators also filtered out questions for qualitative reasons, such as leading to uninformative answers, being too ambiguous to permit a definitive answer, or requiring external knowledge that was unavailable. This human-in-the-loop process yielded a curated set of 211 high-quality QA pairs with 178 datasets. Furthermore, all the questions were rephrased by separating the output format (e.g. bar chart, list of tuples) from the question, enhancing the naturalness, and paraphrasing the column names mentioned in the questions. As a final quality control measure, a second group of annotators with much experience in data science, who were not involved in the initial revision phase, performed a concluding review by using the different annotation GUI as presented in Figure 8. This step was designed to validate the quality and logical soundness of the final QA pairs, with a particular focus on ensuring the Python code was robust and accurately addressed the corresponding question.

After the question scoring stage, we generated a total of 1,840 candidate questions, from which we curated 211 high-quality QA pairs, requiring 9,246 LLM calls in total. A detailed breakdown of the reasons for discarding candidate questions is provided in Appendix B.3.

### 2.1.3 ANNOTATION OF INSIGHT

Establishing a ground truth for insight generation is inherently more complex than for question answering. The subjective nature of what constitutes a meaningful finding makes achieving consensus difficult, posing a significant challenge for both automated generation and evaluation. To address this limitation, we adopted official reports that accompany the open datasets. These human-authored documents contain the key findings and conclusions originally derived by specialists. Our process involved curating six datasets (Table 8) that included such reports. We then systematically extracted the principal findings from each document depending on the representation of the key findings in

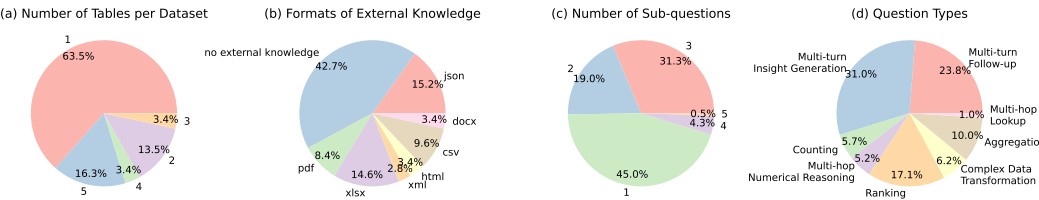

Figure 3: Distributions of (a) number of tables per dataset, (b) format of external knowledge, (c) number of sub-questions, and (d) question types

the report. If the report presents insights as bullet points, we directly treated each bullet point as one insight. If the report expresses findings in free text, we uploaded the report to NotebookLM (Google, accessed July 10, 2025) and prompted it to suggest ten insights from the results sections, followed by the manual verification of the quality of the extracted insights. We then synthesized the extracted insights into a standardized set of declarative sentences, and converted the set of sentences into a summary via Gemini 2.5 Flash (Comanici & et al, 2025). The set of insights and the summary together form the ground-truth corpus for our insight generation task.

## 2.2 BENCHMARK STATISTICS

**Dataset Statistics**: A statistical overview of the datasets in OpenDataBench is presented in Table 2. The benchmark comprises 178 unique datasets, featuring tables with an average of approximately 210K rows and 18 columns. The scale of the data is substantial, with the largest table containing up to 11.9M rows and 213 columns that exceed those found in most existing table QA benchmarks. The distribution of original open data websites is presented in Table 7. Figure 3 (a) illustrates the distribution of tables per dataset; notably, over 36% of the datasets are multi-tabular, with five tables as the most frequent configuration. Furthermore, as shown in Figure 3 (b), over 57% of the datasets are accompanied by external knowledge to aid data interpretation, provided in various formats (e.g. PDF, XLSX). These characteristics—including large, multi-table schemas and the integration of external knowledge—underscore the alignment with realistic data analysis scenarios.

**Task Statistics**: Table 2 provides a statistical summary of tasks in the benchmark. The Table QA task includes 211 question sets, which are categorized into 95 simple questions and 116 decomposable questions. When these decomposable questions are broken down into their constituent parts, the benchmark contains a total of 414 individual questions. The distribution of sub-questions per question set is detailed in Figure 3 (c), which shows that over 55% of all question sets include multiple sub-questions. Figure 3 (d) also presents the distribution of question types. For the Table Insight task, a curated subset of six datasets, each accompanied by a report from domain experts, is designated for insight generation evaluation.

Table 2: Statistics of datasets & tasks

| Properties | Value |
|---|---|
| #Datasets | 178 |
| #Average Rows | 210K |
| #Max Rows | 11.9M |
| #Average Columns | 18.4 |
| #Max Columns | 213 |
| #Datasets for Table QA | 173 |
| #Datasets for Table Insight | 6 |
| #Simple Questions | 95 |
| #Decomposable Questions | 116 |
| #Individual Questions | 414 |

## 3 EXPERIMENTAL SETUP

Our benchmark evaluates performance on two distinct tasks: *Table QA* and *Table Insight*. For Table QA, models receive a user question that specifies the desired output format (text or visualization). In decomposable questions, the preceding conversational history is also provided. The goal is to produce a precise textual or visual answer. The Table Insight task challenges models to generate a list of findings and a summary directly from the given files. Implementation details, including model configurations, hyperparameters, and settings for fair comparison are available in Appendix C.1.

### 3.1 EVALUATION COMPARISONS FOR TABLE QA

We evaluate a range of baselines, from general-purpose LLMs to specialized agents.

Figure 4: Architecture of (a) Answer Agent and (b) Insight Agent

**LLM**: We evaluate a diverse set of LLMs to investigate the capability of table reasoning. We selected open-source models from several categories: general-purpose (Llama 3.1 (Grattafiori et al., 2024), DeepSeek-R1 (DeepSeek-AI et al., 2025), and Qwen3 (Yang et al., 2025)), code generation-specific (Devstral (MistralAI, accessed July 10, 2025) and Qwen3-Coder (Qwen, accessed July 22, 2025)), and table-specific (TableGPT2 (Su et al., 2024)). We also include high-performance closed-source models, namely GPT-4o (OpenAI et al., 2024b) and Gemini-2.5 Flash (Comanici & et al, 2025).

**Answer Agent (Ours)**: We propose the Answer Agent, designed to robustly generate Python code for answering questions. The agent is composed of the following components shown in Figure 4 (a).

- Feature type-specific table serialization: This module first processes the raw tables using the table serialization based on feature types as detailed in Appendix B.2.2. The resulting structured, textual representation of the data is then utilized as input for the subsequent modules.

- Coding: With the serialized table, metadata, external knowledge, and the input question, this module generates Python code to produce an answer. It incorporates a self-correction mechanism: if code execution fails, a subsequent LLM is called to revise the code based on the error message with up to three revision attempts allowed.

- Reflection: This module addresses cases where successfully executed code produces semantically incorrect outputs (e.g., a visualization with no data points or calculation resulting in NaN). A Vision-Language Model (VLM) or Multimodal Large Language Model (MLLM) is employed to analyze visual outputs, while an LLM analyzes text-based results. If an issue is detected, the module triggers a revision loop to refine the code logic, and this can be repeated to three times.

We also performed preliminary experiments with the specialized table agents as baselines, including InfiAgent-DABench (Hu et al., 2024) and tablegpt-agent (Su et al., 2024); however, their performance on our benchmark was near-zero due to the complexity of our benchmark, so they were excluded from the final comparison.

## 3.2 EVALUATION COMPARISONS FOR TABLE INSIGHT

For the Table Insight task, we evaluate the following baseline agents. We employ only closed-source LLMs in these agents, given the relatively low performance of open-source models on Table QA.

**AgentPoirot** (Sahu et al., 2025): This agent is designed for goal-oriented insight generation. It operates by first extracting the data schema and then generating a set of high-level questions. For each question, it generates an answer, interprets it, and recursively poses follow-up questions to dive deeper, and finally summarizes the obtained insights. This process follows a tree-like exploration structure (Figure 9 (a)), where each branch represents a deep dive into a specific analytical path.

**Insight Agent (Ours)**: We propose the Insight Agent shown in Figure 4 (b), a novel framework that iteratively generates questions, produces answers, and derive insights. The agent begins by generating the fixed number of high-level questions, which are then processed by our Answer Agent to obtain correct answers. Insights are subsequently synthesized from multiple QA pairs. These initial insights then seed the generation of new follow-up questions by combining multiple insights, continuing the cycle, ending by the summarization. Unlike AgentPoirot's tree-structured approach, the Insight Agent employs a directed acyclic graph (DAG)-based approach as explained in Figure 9 (b), as the generation of new questions and insights selects and aggregates the context from all previously generated information instead of single insight or QA pair in the previous depth, as explained in Appendix C.2.

## 3.3 EVALUATION METRICS

To assess the performance of agents and LLMs on our benchmark, we compare their outputs against the ground-truth references. Distinct evaluation protocols are employed for Table QA and Table Insight tasks.

**Table QA**: The QA task is evaluated on accuracy under two settings: *Whole*, where all sub-questions in a decomposable question must be correct, and *Individual*, which measures sub-question-level accuracy. Correctness is determined by the modality of the answer. Text-based answers are judged by Exact Match (EM). Visualizations are evaluated using an MLLM-as-a-judge protocol, where four MLLMs (GPT-4o, GPT-4o-mini, Gemini 2.5 Flash, and Gemini 2.5 Pro (Comanici & et al, 2025)) assess the semantic equivalence between the predicted and ground-truth outputs. The judges are provided both the rendered images and their source code, and a prediction is deemed correct upon a majority consensus, requiring positive assessments from at least three of the four models.

**Table Insight**: To evaluate the quality of generated insights, we adopt the methodology from Insight-Bench (Sahu et al., 2025) computing LLaMA-3-Eval scores. We employ GPT-4o as the evaluator by replacing LLaMA3-70b (Grattafiori et al., 2024). The evaluation is conducted at two levels of granularity:

- Summary-level Score: This metric assesses the holistic quality of the entire set of generated insights by comparing it against the complete ground-truth summary.
- Insight-level Score: This metric provides a more fine-grained analysis. It measures the semantic alignment between each individual ground-truth insight and the most relevant insight from the predicted set, with the final score being the average of these individual comparisons.

## 3.4 IMPLEMENTATION DETAILS

The proposed Answer Agent requires a VLM or a MLLM within its Reflection Module to validate visual outputs. For experiments involving closed-source models, we utilized their native multimodal capabilities across all modules. For the open-source agent configurations, we paired various LLMs with a specialized VLM, Chart-R1-7B (Chen et al., 2025). For the Table Insight task, each experiment was executed five times per model, and the scores were averaged across these runs to ensure the stability of our result. All results were obtained with the model temperature set to 0.0 to promote deterministic output. The other details about LLM configurations and hyperparameters are available in Appendix C.1.

## 4 EVALUATION RESULTS

### 4.1 META EVALUATION OF TABLE INSIGHT

We conducted a meta evaluation to assess the validity of the evaluation metrics for Table Insight. We sampled 50 pairs of generated insights (Insight A and Insight B) for each ground-truth insight and asked three independent annotators to assess which of the two was closer to the ground truth. Annotators selected one of five relative options: **A+** (*A is definitely better*), **A** (*A is slightly better*), **N** (*A and B are comparable*), **B** (*B is slightly better*), or **B+** (*B is definitely better*). Each option was mapped to a normalized score in [-2, -1, 0, 1, 2]. The final human-judgment score for each pair was obtained by averaging the three annotators' scores. In parallel, we computed a metric-derived score based on the difference between the insight-level scores of B and A. We then quantified the agreement between human judgments and the metric-derived scores using both Pearson and Spearman correlations, which yielded coefficients of 0.669 and 0.663, respectively. These results indicate a solid alignment between the proposed metric and human judgments, providing empirical support for the reliability of our evaluation methodology for the Table Insight task.

### 4.2 QUANTITATIVE EVALUATION RESULTS

The main results for the Table QA and Table Insight tasks are presented in Table 3, where *w/o Answer Agent* means that the LLM is executed to generate the Python codes once based on the first 10 rows of the tables instead of the specific serialization. For the Table QA task, our Answer Agent

consistently outperforms the base LLMs. With Gemini 2.5 Flash, for instance, it achieves relative improvements of approximately 27% in the Whole setting and 25% in the Individual setting. This suggests that a structured agentic framework is crucial, as standalone LLM reasoning is insufficient for such complex tasks. Despite these gains, the top absolute score in the Whole setting remains below 0.4, highlighting the difficulty of the benchmark. Among the open-source models, Qwen3-30B achieves the highest score in the Whole setting, while Devstral-small performs best in the Individual setting. However, their performance still lags behind that of the closed-source models. Notably, TableGPT2-7B, a model specialized for tabular data, scores below 0.1 even when paired with our Answer Agent.

In the Table Insight task, our Insight Agent outperforms the AgentPoirot baseline, achieving relative improvements of 11% (insight-level) and 13% (summary-level) with Gemini 2.5 Flash. This superiority holds at the dataset level (Table 10), where our agent wins on a majority of datasets for both insight-level (5 out of 6) and summary-level (4 out of 6) scores. However, the absolute scores remain low even with the top-performing model, indicating substantial challenges remain in automated insight generation.

### 4.3 QUALITATIVE ANALYSIS

#### 4.3.1 ERROR ANALYSIS OF TABLE QA

We conducted an error analysis on the incorrect answers (the Individual setting) from Gemini 2.5 Flash with Answer Agent, with the results categorized in Figure 5.

The most prevalent issue, **Condition Filter Error (32.4%)**, occurs when the model fails to apply implicit conditions not explicitly stated in the question. A common example involves datasets with aggregated and disaggregated data (e.g., population counts for 'male', 'female', and 'total'); models often fail to apply appropriate filters to avoid double-counting, leading to incorrect calculations. The second most frequent category is **Transformation Error (23.2%)**, which involves failures in data wrangling and type conversion. Common mistakes include parsing-related failures various datetime formats (e.g. day-first format) by using `pandas.to_datetime` method or neglecting to convert numerical strings (e.g., "1,234,567") into integer or float types. To mitigate these errors, the more comprehensive yet efficient view of tables (e.g. exploratory data analysis results) is required in addition to the feature type-based representations for future work. Additionally, integrating SQL-based operations into the code generation process would be beneficial because SQL offers strong capabilities for structured data manipulation, while Python remains more flexible for tasks such as visualization and advanced statistical analysis.

Errors also arise from the inherent complexity of the tasks. **Context Handling Errors (8.7%)** occur in decomposable questions where the model incorrectly uses the output from a flawed previous turn though the generated logic is correct in most cases. **Visualization Errors (6.8%)** typically involve incorrect axis ranges, such as a timeline that does not match the period specified in the question. Finally, the complexity of the benchmark's data structure leads to specific errors. These include **Wrong Choice of Tables (4.3%)** in multi-table scenarios. This error occurs when the generated code fails to select the correct table from a multi-table dataset based on information provided in the metadata. For example, a dataset may contain separate tables for annual statistics, with the year covered by each table specified only in the metadata. An error arises if a question pertains to a specific year, but the model fails to refer to the metadata and consequently queries the wrong table. **Misunderstanding External Knowledge (3.9%)**, where the model fails to correctly apply information from provided data dictionaries to interpret the data.

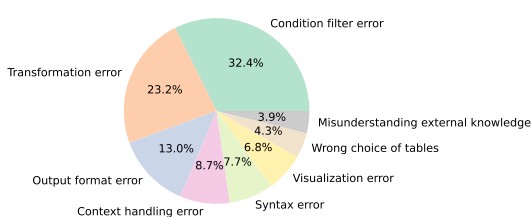

Figure 5: Error distribution with Answer Agent

| LLM | Table QA | | | | Table Insight | | | |
|---|---|---|---|---|---|---|---|---|
| | w/o Answer Agent | | w/ Answer Agent | | AgentPoirot | | Insight Agent | |
| | Whole | Individual | Whole | Individual | Insight | Summary | Insight | Summary |
| **Closed-source LLMs** | | | | | | | | |
| Gemini 2.5 Flash | 0.310 | 0.401 | **0.393** | **0.502** | 0.283 | 0.359 | 0.315 | **0.405** |
| GPT-4o | 0.242 | 0.333 | 0.270 | 0.391 | 0.292 | 0.345 | **0.319** | 0.399 |
| **Open-source LLMs** | | | | | | | | |
| Qwen3-30B | 0.134 | 0.216 | 0.199 | 0.309 | | | | |
| Qwen3-Coder-30B | 0.123 | 0.191 | 0.186 | 0.280 | | | | |
| Devstral-Small | 0.152 | 0.221 | 0.186 | 0.316 | | | | |
| DeepSeek-R1-14B | 0.038 | 0.061 | 0.095 | 0.140 | | | | |
| Llama3.1-8B | 0.019 | 0.017 | 0.019 | 0.034 | | | | |
| TableGPT2-7B | 0.057 | 0.088 | 0.066 | 0.109 | | | | |

Table 3: Main results of Table QA and Table Insight with LLMs and specific agents. *w/o Answer Agent* is without the agentic support.

### 4.3.2 FINE-GRAINED ANALYSIS OF TABLE INSIGHT

While the evaluation metrics provides a single value to measure the alignment between predicted and target insights, we conduct a more fine-grained analysis to understand specific model capabilities. We decompose the evaluation into four distinct perspectives: **Topic Relevance** (*Does the predicted insight address the same topic as the target?*), **Narrative Alignment** (*Does the prediction make the same core argument or conclusion as the target?*), **Qualitative Details Match** (*Does the prediction mention the same specific names or entities as the target?*), and **Quantitative Details Match** (*Does the prediction mention the same specific quantitative values as the target?*). For each of the 280 pairs [1], we prompted GPT-4o to score the prediction on each perspective using a 1–5 scale. The score distributions for both Insight Agent and AgentPoirot are presented in Figure 6.

The results show that both agents perform relatively well on Topic Relevance, the highest-level criterion. However, for Qualitative Details Match and Narrative Alignment, both agents struggle to achieve high scores, though our Insight Agent outperforms AgentPoirot in these scores. This indicates our agent is more capable of drawing correct conclusions and referencing specific entities, a finding supported by the qualitative examples in Table 13. Finally, both agents fail on Quantitative Details Match, with none of the predicted insights scoring 3 or higher. This highlights an essential area for future work: improving the ability of agents to accurately calculate and present specific numerical values in their generated insights.

### 4.4 ABLATION STUDY

To assess the contribution of each Answer Agent module, we conducted an ablation study against a naive baseline using the first 10 table rows (Table 4) for the Python code generation. Adding the table schema consisting of column names, data types, and statistics on missing and unique values does not affect the performance. We further increased the number of provided rows whose number was increased up to a maximum of 200, adjusted as needed to fit within the LLM context window. The performance degraded noticeably, suggesting that LLMs struggle to identify the key information needed for correct solution construction when faced with a larger pool of numerical or textual entries, compared to the first 10 rows. In contrast, the proposed feature type-specific serialization led to substantial gains over the baseline by providing a compact and reliable representation of the table that avoids the baseline's tendency to rely on guessed values. Adding the Reflection module further improved performance across models by enabling the system to capture implicit constraints and infer missing reasoning steps.

Table 4: Ablation Study of Answer Agent

| Settings | Gemini 2.5 Flash | GPT-4o |
|---|---|---|
| Baseline | 0.310 | 0.242 |
| + Schema | 0.308 | 0.244 |
| + Schema + More number of rows | 0.275 | 0.232 |
| + Serialization | 0.360 | 0.257 |
| + Serialization + Reflection | 0.379 | 0.267 |
| + Serialization + Reflection + Self-correction (Answer Agent) | 0.393 | 0.270 |

---

[1] 5 executions × 56 GT insights (10 per 5 datasets, 6 for 1 dataset)

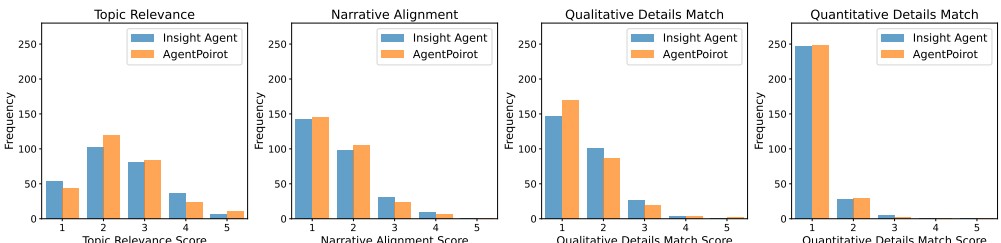

Figure 6: Comparison of insight perspective distributions between Insight Agent and AgentPoirot.

## 5   RELATED WORKS

**Benchmarks for Table Question and Answering**.  Research in table-based question answering has been largely driven by a series of influential benchmarks. Early work such as WikiTableQuestions (Pasupat & Liang, 2015) established the task of answering natural language questions over Wikipedia tables. While foundational, this benchmark is limited to single tables and relatively simple questions. Although benchmarks like HybridQA (Chen et al., 2020), FeTaQA (Nan et al., 2022), and OTT-QA (Chen et al., 2021) introduced tasks requiring more complicated reasoning, they still focus on small-scale Wikipedia tables. The subsequent development of text-to-SQL benchmarks marked a significant leap in complexity. Spider (Yu et al., 2018) and BIRD (Li et al., 2023) became the standards requiring models to generate complex SQL queries. More recently, developments in LLMs have enabled models to generate coherent Python code, leading to the construction of benchmarks that assess data analysis capabilities (Wu et al., 2025b; Hu et al., 2024; Osés Grijalba et al., 2024). While these benchmarks were instrumental in advancing table reasoning capabilities, they do not comprehensively capture the challenges of real-world data analysis. The datasets are typically well-structured and moderate in scale. They often lack key practical characteristics, such as rich metadata and supplementary external knowledge. Furthermore, many do not address the large multi-tabular datasets. OpenDataBench comprehensively targets these limitations by incorporating all of these features: large tables, multi-table schemas, metadata, and extenral knowledge.

**Benchmarks for Table Insight Generation**.  Automated insight generation is a nascent and challenging area to benchmark, as the subjective nature of an "insight" complicates objective evaluation (Zhang & Elhamod, 2025; Majumder et al., 2024).  Recent work like InsightBench (Sahu et al., 2025) has advanced this area by using LLM-based evaluation to score findings from table data with small variations (ServiceNow tables). However, this approach has limitations: its underlying datasets lack the diversity and actual scale of public data, and its proposed agent, AgentPoirot, employs a tree-based exploration without the reflection component, which can constrain the diversity and correctness of its findings. In contrast, OpenDataBench provides a benchmark with diverse and large-scale tables. Furthermore, our proposed agent architecture is explicitly designed to improve correctness and diversity through integrated verification modules and a more flexible DAG-based exploration process.

## 6   CONCLUSION

To address the critical lack of realism in existing benchmarks, we introduced *OpenDataBench*, a new benchmark built from open data. It features large, multi-tabular datasets, and incorporates external knowledge, and formalizes two key tasks: complex Question Answering (with decomposable questions and visualizations) and a novel Insight Generation task grounded in reports by domain specialists. Our extensive evaluation reveals that even state-of-the-art models struggle with low QA accuracy. While our proposed *Answer Agent* and *Insight Agent* improve upon baselines, their performance still highlights the difficulty of these tasks. A detailed qualitative analysis provides a clear roadmap for future research. Future efforts should focus on both addressing these identified issues and efficiently scaling the benchmark's size to ensure more robust evaluations. We believe OpenDataBench will serve as a catalyst steering research toward building more robust agents capable of handling real-world data complexities.

## REPRODUCIBILITY STATEMENT

We provide code, sampled datasets, and software dependencies in the anonymous GitHub repository. Implementation details (infrastructure, hyperparameters, LLM models, and fair-comparison settings) and evaluation protocols are described in Section 3, Appendix C.1 and source codes. Prompts for benchmark construction and agents are provided in Appendix E. The benchmark construction procedure and list of data sources are given in Section 2.1 and Appendix B.

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

## A  USE OF LARGE LANGUAGE MODELS

We used Large Language Model (LLM) for the grammar correction and the words refinement to enhance the quality of the paper.

## B  BENCHMARK CONSTRUCTION

**The complete benchmark will be publicly released with the camera-ready version.**

### B.1  DATA CURATION

#### B.1.1  DATA COLLECTION

Table 5 lists 53 websites, from which we downloaded all datasets along with their metadata via APIs (e.g., CKAN API [2]).

#### B.1.2  DATA FILTERING

All downloaded datasets were subjected to a rigorous filtering and curation process to select those most suitable for real-world data analytics tasks. This process involved three main stages: dataset filtering, external knowledge identification, and metadata standardization. First, data filtering is conducted based on the following conditions:

- The dataset had to contain at least one CSV file that was correctly formatted and readable by the `pandas.read_csv` method. Files that were HTML or XML in content despite having a .csv suffix were excluded.

- At least one CSV file within the dataset was required to have more than 5,000 rows. Additionally, tables with five or more blankc columns were discarded.

- Each dataset needed to be accompanied by a textual description. The license was also required to permit redistribution; for datasets from Data.gov where the license was often unspecified in the metadata, we performed manual verification on the source webpage.

- ArcGIS-based datasets, which are primarily geospatial, were excluded from our analysis.

Following the filtering stage, we systematically searched for external knowledge (e.g., data dictionaries) within each dataset using a set of heuristic rules:

- First, an automated search was performed for files with names containing "data dictionary" or "datadictionary".

- Next, a platform-specific rule was applied for Data.gov datasets. We observed that when a JSON file is provided alongside CSV, XML, and RDF files, it often contains column-level descriptions. In such cases, the JSON file was designated as external knowledge.

- If these automated heuristics failed, we performed a manual inspection of the dataset's contents to locate any other supplementary documentation that could serve as a data dictionary.

Finally, the original metadata for each curated dataset was processed and standardized. This step created a concise metadata format specific to our benchmark by removing redundant or irrelevant information from the source. The following is an example of the specific metadata from *Indiana Arrest Data* of *Indiana Data Hub*.

Converted Metadata

```
"identifier": "d39f6598-efbb-40a7-a694-6a9b8d2dc2dc"
"dataset_title": "INDIANA ARREST DATA"
```

---

[2]`https://github.com/ckan/ckanapi`

```
972    "dataset_description": "This dataset is the underlying data of the
973        Indiana Arrests Dashboard which displays counts of individuals
974        arrested, arrests, charges by offense category, dispositions, country
975         and time period in Indiana beginning in 2008 through the present
976        year. \r\n\r\nArrest data comes from the Criminal History Repository
977        System (CHRIS). Data feeding into the CHRIS system comes from three
978        main sources. Arrest data comes from the LiveScan system, which is
979        used for fingerprinting and capturing other pertinent information at
980        the time of the arrest. Criminal disposition data are maintained by
981        prosecutors in ProsLink system, and by the courts in the Odyssey
982        system. \r\n\r\nData Notes:\r\n\r\n1. Arrest data are sent to ISP
983        soon after the arrest occurs, but disposition data have a lag of
984        approximately seven months as the case makes its way through the
985        legal system. \r\n\r\n2. Text descriptions of the original offenses
986        are provided by the arresting officer when the offender is arrested.
987        Later, the prosecutor's office or court provides a text description
988        of the filed offense, along with the Indiana Code title, article,
989        chapter, and section (e.g.35-48-4-6). The filed offense may be
990        amended later. \r\n\r\n3. Arrest County is determined by the location
991         of the booking agency. If the booking agency is missing, then the
992        arresting agency is used. \r\n\r\n4. The count of individuals/arrests
993        /charges by offense category can add up to more than the grand total
994        because one individual/arrest/charge can fall into multiple
995        categories (e.g. DUI is counted in the \"Drug\" and \"Traffic\"
996        categories. \r\n\r\n5. Arrest categories and subcategories are
997        determined based on keywords found in a free text description of the
998        offense. About 7% of offenses have a description that has not yet
999        been categorized."
996    "publisher": "Indiana State Police"
997    "landingPage": "Indiana State Police"
998    "license": "Creative Commons Attribution"
999    "distribution": [{"file_name": "data9.csv",
1000              "file_title": "ARREST DATA 2022 Q3",
1001              "file_description": null,
1002              "downloadURL": "https://hub.mph.in.gov/dataset/d39f6598-efbb
1003                  -40a7-a694-6a9b8d2dc2dc/resource/00cd698d-e26b-458a-861b
1004                  -4c355b77ab20/download/isp_arrest_data_2022_q3.csv",
1005              "accessURL": "https://hub.mph.in.gov/dataset/d39f6598-efbb
1006                  -40a7-a694-6a9b8d2dc2dc/resource/00cd698d-e26b-458a-861b
1007                  -4c355b77ab20/download/isp_arrest_data_2022_q3.csv"},
1008            {"file_name": "data37.csv",
1009             "file_title": "ARREST DATA 2015 Q3",
1010             "file_description": null,
1011             "downloadURL": "https://hub.mph.in.gov/dataset/d39f6598-efbb
1012                 -40a7-a694-6a9b8d2dc2dc/resource/8b2b54fe-363a-46f7-9c3b
1013                 -197cce01616f/download/isp_arrest_data_2015_q3.csv",
1014             "accessURL": "https://hub.mph.in.gov/dataset/d39f6598-efbb
1015                 -40a7-a694-6a9b8d2dc2dc/resource/8b2b54fe-363a-46f7-9c3b
1016                 -197cce01616f/download/isp_arrest_data_2015_q3.csv"},
1017            {"file_name": "data20.csv",
1018             "file_title": "ARREST DATA 2019 Q4",
1019             "file_description": null,
1020             "downloadURL": "https://hub.mph.in.gov/dataset/d39f6598-efbb
1021                 -40a7-a694-6a9b8d2dc2dc/resource/bd011a33-0652-4ad7-8d90
1022                 -6c1019d6385c/download/isp_arrest_data_2019_q4.csv",
1023             "accessURL": "https://hub.mph.in.gov/dataset/d39f6598-efbb
1024                 -40a7-a694-6a9b8d2dc2dc/resource/bd011a33-0652-4ad7-8d90
1025                 -6c1019d6385c/download/isp_arrest_data_2019_q4.csv"},
              {"file_name": "data15.csv",
               "file_title": "ARREST DATA 2021 Q1",
               "file_description": null,
               "downloadURL": "https://hub.mph.in.gov/dataset/d39f6598-efbb
                   -40a7-a694-6a9b8d2dc2dc/resource/9c7960c6-417b-45e6-9ace
                   -b75958dd91de/download/isp_arrest_data_2021_q1.csv",
```

```
                  "accessURL": "https://hub.mph.in.gov/dataset/d39f6598-efbb
                     -40a7-a694-6a9b8d2dc2dc/resource/9c7960c6-417b-45e6-9ace
                     -b75958dd91de/download/isp_arrest_data_2021_q1.csv"},
               {"file_name": "data14.csv",
                "file_title": "ARREST DATA 2021 Q2",
                "file_description": null,
                "downloadURL": "https://hub.mph.in.gov/dataset/d39f6598-efbb
                     -40a7-a694-6a9b8d2dc2dc/resource/1ff2cf5f-69ef-4139-bcb4
                     -036f66787172/download/isp_arrest_data_2021_q2.csv",
               "accessURL": "https://hub.mph.in.gov/dataset/d39f6598-efbb
                     -40a7-a694-6a9b8d2dc2dc/resource/1ff2cf5f-69ef-4139-bcb4
                     -036f66787172/download/isp_arrest_data_2021_q2.csv"}],
"external_knowledge": ["data68.xlsx"]}
```

## B.2    DATA ANNOTATION

### B.2.1    QUESTION TYPES

During question generation, specific question types are provided to the LLM to guide the formulation of questions. We use the following eight question types, with each type's name and description supplied to the LLM. *Multi-turn Follow-up* and *Multi-turn Insight Generation* correspond to decomposable questions.

- **Aggregation**: Questions involving aggregated answers based on the statistical operations, such as mean, sum or mode

- **Ranking**: Questions involving answers based on the ranking

- **Counting**: Questions involving answers based on counting something

- **Multi-hop Lookup**: Question involving extracting single cell value from the table based on multiple reasoning steps

- **Multi-hop Numerical Reasoning**: Questions involving numerical answers based on multiple reasoning steps

- **Complex Data Transformation**: Question involving complex data transformation, such as aggregation or filtering across multiple dimensions, creating new columns, filtering with context-dependent logic, resolving entity references across rows, or merging multiple tables

- **Multi-turn Follow-up**: Question involving multi-turn follow-up questions that build on previous answers or context from table data, requiring the model to maintain state and context across multiple interactions

- **Multi-turn Insight Generation**: Question involving multi-turn insight generation that requires the model to generate insights or summaries based on previous answers or context from table data, requiring the model to maintain state and context across multiple interactions. Questions in the intermediate turn ask to provide not only text-based answer but also text-based complicated statistical information (e.g. correlation) and visualization

### B.2.2    FEATURE TYPE-SPECIFIC TABLE SERIALIZATION

Our serialization process generates a compact textual representation of a table by summarizing its global properties and providing detailed, feature type-aware information for each column. The serialized text begins with the dimensions of the tables (number of rows and columns), followed by a per-column breakdown. For each column, the serialization includes: the inferred feature type, the Pandas data type (Wes McKinney, 2010), the percentage of NaN values, and a feature-specific textual summary. The primary feature type is determined by a Feature Type Inference (FTI) model (Liu et al., 2024), which is based on a trained Random Forest Classifier. This model classifies each column into one of 11 types: *Numerical*, *Categorical*, *Datetime*, *Sentence*, *URL*, *Embedded Number*, *List*, *Ignorable ID*, *Numbers with Unit*, *Numbers with Sign*, *Range of Numbers*, or *Formatted ID*. The Pandas data type is inferred using the built-in `pandas.api.types.infer_dtype` function. While its output would overlap with the feature types, we include it because its ability to

Table 5: List of Open Data Websites

| Websites | URL |
|---|---|
| Data.gov | https://data.gov/ |
| California Open Data Portal | https://data.ca.gov/ |
| Hawaii Open Data | https://opendata.hawaii.gov/ |
| Analyze Boston | https://data.boston.gov/ |
| City of Houston Open Data | https://data.houstontx.gov/ |
| The Indiana Data Hub | https://hub.mph.in.gov/ |
| Milwaukee Open Data | https://data.milwaukee.gov/ |
| Open Data SA | https://data.sanantonio.gov/ |
| Pompano Beach Open Data | https://data.pompanobeachfl.gov/ |
| America's Education data | https://data.ed.gov/ |
| Energy Data eXchange | https://edx.netl.doe.gov/ |
| California Health and Human Services Open Data Portal | https://data.chhs.ca.gov/ |
| California Natural Resources Agency Open Data | https://data.cnra.ca.gov/ |
| U.S. Small Business Administration Open Data | https://data.sba.gov/ |
| Ireland's Open Data Portal | https://data.gov.ie/ |
| Dublinked: Open Data for the Dublin Region | https://data.smartdublin.ie/ |
| Tusla Data Catalogue | https://datacatalog.tusla.ie/ |
| DAFM Data Portal | https://opendata.agriculture.gov.ie/ |
| Central Bank of Ireland's Open Data Portal | https://opendata.centralbank.ie/ |
| Data.gov.au | https://data.gov.au/ |
| The Central Resource for SEED in NSW | https://www.seed.nsw.gov.au/ |
| Data.NSW | https://data.nsw.gov.au/ |
| NTG Open Data Portal | https://data.nt.gov.au/ |
| Data.SA | https://data.sa.gov.au/ |
| Ballarat Open Data | https://ballaratopendata.org.au/ |
| DATA VIC | https://www.data.vic.gov.au/ |
| Data WA | https://www.data.wa.gov.au/ |
| Queensland Government Publications Portal | https://www.publications.qld.gov.au/ |
| Transport Open Data | https://opendata.transport.nsw.gov.au/ |
| openAFRICA | https://open.africa/ |
| Data.gov.hk | https://data.gov.hk/en/ |
| Data.gov.uk | https://www.data.gov.uk/ |
| UK Data Service | https://statistics.ukdataservice.ac.uk/ |
| London Datastore | https://data.london.gov.uk/ |
| Open Data NI | https://admin.opendatani.gov.uk/ |
| ENTSO-E | https://docs.entsoe.eu/ |
| Journal Data Archive | https://journaldata.zbw.eu/ |
| Data.openstate.eu | https://data.openstate.eu/ |
| OPERANDUM | https://data-catalogue.operandum-project.eu/ |
| Dataportal.ponderful.eu | https://dataportal.ponderful.eu/ |
| OpenCity | https://opencity.in/ |
| New Zealand's Biological Heritage Data Repository | https://data.bioheritage.nz/ |
| Datastore.landcareresearch.co.nz | https://datastore.landcareresearch.co.nz/ |
| Open.canada | https://search.open.canada.ca/opendata/ |
| Open Govermental Portal in Alberta | https://www.alberta.ca/open-government-program |
| Data.gov.bc.ca | https://catalogue.data.gov.bc.ca/ |
| Niagara's Open Data Catalogue | https://niagaraopendata.ca/ |
| Ontario Data Catalogue | https://data.ontario.ca/ |
| Données Québec | https://www.donneesquebec.ca/ |
| Surrey's Open Data | https://data.surrey.ca/ |
| City of Toronto's Open Data Portal | https://open.toronto.ca/ |
| Data.sustain.ubc.ca | https://data.sustain.ubc.ca/ |
| Columbia Basin Water Hub | https://data.cbwaterhub.ca |

identify mixed types (e.g., columns containing both strings and integers) serves as a key signal for potential data quality issues that would require wrangling.

The feature type-specific summary is constructed according to the inferred feature type, as follows:

- Numerical: The minimum and maximum values in the column are included.
- Categorical: If the column contains 20 or fewer unique categories, all are listed. Otherwise, a random sample of 20 unique categories is provided.
- Datetime: The earliest and latest date or time values are included.
- URL: No sample values are included. This is a deliberate choice to conserve context length, as full URLs are token-intensive and typically have low semantic value for general data analysis tasks.
- All Other Types: For all other feature types, a random sample of 10 unique values is included to provide a representative snapshot of the column's contents.

The example of the serialized text is provided in the following from *E-bike Field Study* of *Data.gov*.

Text Example by Feature type-specific Table Serialization

```
1st table
Dataset title: Comma Separated Values File
Dataset description: None

Headers and values:
Number of columns: 21
Number of rows: 408363

Feature type, pandas type, ratio of missing values, and feature type-
    specific information is given for each column as below.

date (feature type: Datetime) (pandas type: string) (ratio of missing
    values: 0%): Start date is 2022-04-27 23:42:29.834000+00:00, and end
    date is 2022-09-23 18:31:05.502000+00:00.
lat (feature type: Numerical) (pandas type: floating) (ratio of missing
    values: 0%): Value range is [42.447303, 42.461437200000006].
lon (feature type: Numerical) (pandas type: floating) (ratio of missing
    values: 0%): Value range is [-71.3243906, -71.2562746].
spd (feature type: Numerical) (pandas type: floating) (ratio of missing
    values: 0%): Value range is [0.0, 23.825000000000003].
blind_turn (feature type: Categorical) (pandas type: integer) (ratio of
    missing values: 0%): All categories are [0, 1].
constrained_tunnel (feature type: Categorical) (pandas type: integer) (
    ratio of missing values: 0%): All categories are [0, 1].
narrow (feature type: Categorical) (pandas type: integer) (ratio of
    missing values: 0%): All categories are [0, 1].
slow_sign (feature type: Categorical) (pandas type: integer) (ratio of
    missing values: 0%): All categories are [0, 1].
trail_hazards (feature type: Categorical) (pandas type: integer) (ratio
    of missing values: 0%): All categories are [0, 1].
trail_junction (feature type: Categorical) (pandas type: integer) (ratio
    of missing values: 0%): All categories are [0, 1].
vehicle_conflict_point (feature type: Categorical) (pandas type: integer)
    (ratio of missing values: 0%): All categories are [0, 1].
walk_bike_sign (feature type: Categorical) (pandas type: integer) (ratio
    of missing values: 0%): All categories are [0, 1].
eb (feature type: Categorical) (pandas type: integer) (ratio of missing
    values: 0%): All categories are [0, 1].
uphill (feature type: Categorical) (pandas type: integer) (ratio of
    missing values: 0%): All categories are [0, 1].
downhill (feature type: Categorical) (pandas type: integer) (ratio of
    missing values: 0%): All categories are [0, 1].
passing (feature type: Categorical) (pandas type: integer) (ratio of
    missing values: 0%): All categories are [0, 1].
participantid (feature type: Numerical) (pandas type: integer) (ratio of
    missing values: 0%): Value range is [1, 37].
```

```
age (feature type: Numerical) (pandas type: integer) (ratio of missing
    values: 0%): Value range is [27, 65].
sex (feature type: Categorical) (pandas type: string) (ratio of missing
    values: 0%): All categories are [female, male].
bike_type (feature type: Categorical) (pandas type: string) (ratio of
    missing values: 0%): All categories are [conventional, electric].
ebike_class (feature type: Categorical) (pandas type: floating) (ratio of
     missing values: 56%): All categories are [1.0, 2.0, 3.0].
```

### B.2.3 HUMAN VERIFICATION

Figure 7 shows the annotation GUI, built with Streamlit [3], for revising questions and the Python code used to generate answers. Annotators can refer to the LLM-generated answers and code, as well as the underlying tables, metadata, and external knowledge. Figure 8 presents the GUI for reviewing revised QA pairs, where annotators select one of three options——*Good*, *Ambiguous*, or *Wrong Answer*——and may leave comments for the latter two. In total, we obtained 211 datasets, with their original website distribution shown in Table 7. Six of these datasets (Table 8) are used for the Table Insight task.

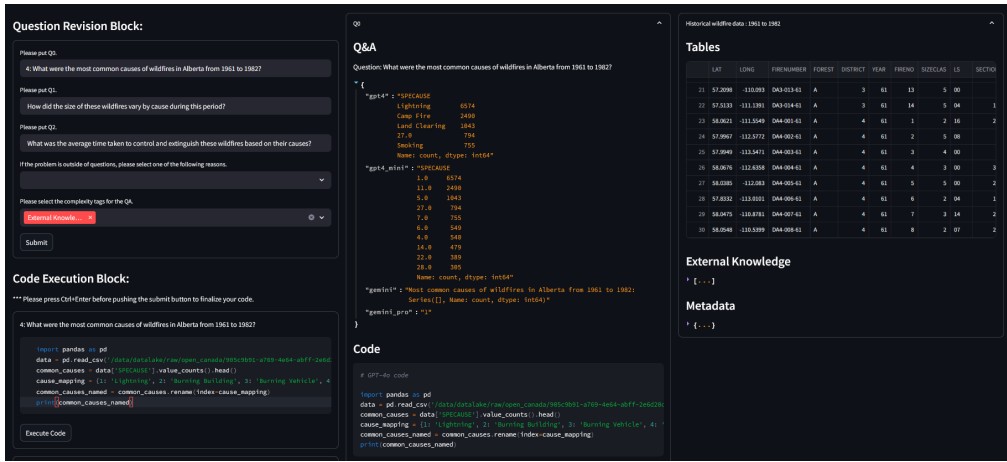

Figure 7: Annotation GUI for revising questions and answers

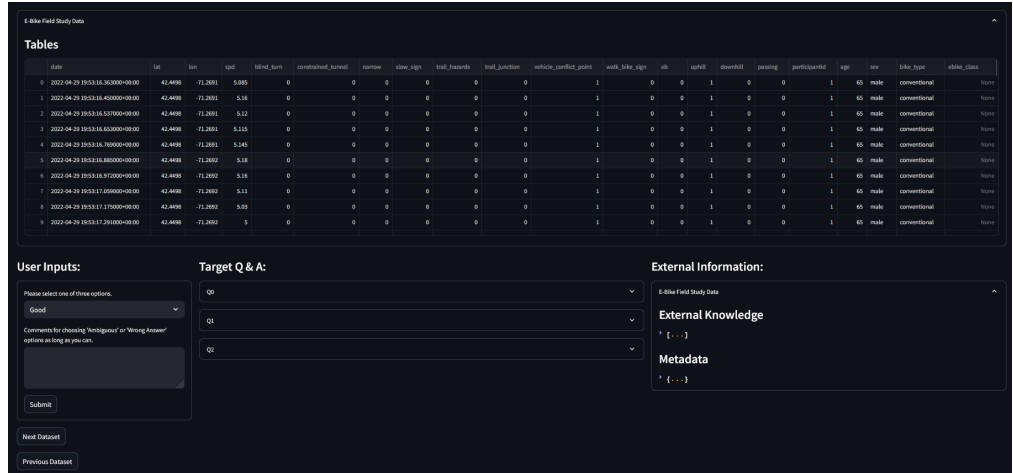

Figure 8: Annotation GUI for checking the revised QA pairs

---

[3]https://github.com/streamlit/streamlit

Table 6: Distribution of Discarding Reasons

| Reason | Count and Ratio |
|---|---|
| Unanimous Agreement | 114 (0.062) |
| Insufficient External Knowledge | 550 (0.299) |
| Ambiguous Question | 354 (0.192) |
| Uninsightful Question | 611 (0.332) |

### B.3 DISTRIBUTION OF DISCARDED CANDIDATE QUESTIONS

We generated 1,840 candidate questions after the question scoring stage and curated them into 211 high-quality questions, discarding the remaining 1,629 based on the criteria below.

- **Unanimous Agreement**: During the answer generation stage, we employed four LLMs to produce answers and measured consensus across models. Questions for which all LLMs produced identical answers were removed because they typically require only shallow reasoning and do not align with the intended complexity of our benchmark. Answer consensus was determined using exact string match for text-based answers and manual verification for visualization-based answers using the GUI tool shown in Figure 7.

- **Insufficient External Knowledge**: Some datasets require external knowledge (e.g. data dictionary) to interpret column meanings or specific values. When such information was missing or insufficient, it became impractical to map column names and values to the generated questions, making accurate answer generation infeasible. Questions of this type were removed.

- **Ambiguous Question**: Questions allowing multiple plausible answers were removed to ensure benchmark clarity. For example, when a table contains both calendar-year and fiscal-year columns, a question asking for "the year" satisfying certain conditions becomes ill-posed unless the question explicitly specifies which type of year should be used.

- **Uninsightful Question**: We excluded questions that failed to yield analytically meaningful or practically useful insights despite being answerable. A common example occurs in geospatial datasets, where questions such as "What is the average latitude and longitude under certain conditions?" often result in a coordinate that lacks geographic or analytical relevance (e.g., a point in the ocean). Such questions were deemed non-insightful and removed.

The distribution of the reasons is shown in Table 6.

Table 7: Distribution of Open Data Websites in OpenDataBench

| Websites | Count |
|---|---|
| Open.canada | 63 |
| Data.gov | 35 |
| California Open Data Portal | 27 |
| Open Govermental Portal in Alberta | 7 |
| Data.gov.uk | 6 |
| Analyze Boston | 5 |
| Ontario Data Catalogue | 4 |
| The Indiana Data Hub | 4 |
| Data.SA | 4 |
| Surrey's Open Data | 4 |
| Open Data NI | 4 |
| The Central Resource for SEED in NSW | 2 |
| Pompano Beach Open Data | 2 |
| Data.NSW | 2 |
| U.S. Small Business Administration Open Data | 1 |
| Hawaii Open Data | 1 |
| City of Houston Open Data | 1 |
| openAFRICA | 1 |
| City of Toronto's Open Data Portal | 1 |
| Milwaukee Open Data | 1 |
| OpenCity | 1 |
| DATA VIC | 1 |
| Columbia Basin Water Hub | 1 |

Table 8: Datasets for Table Insight

| Dataset | Website | Domain |
|---|---|---|
| Boston Buildings Inventory | Analyze Boston | Real Estate |
| Number of Weight Loss Surgeries Performed in California Hospital | Data.gov | Healthcare |
| Cross-Canada Survey of Radon Concentrations in Homes | Open.canada | Environment |
| Fixed gear sentinel fisheries program - northern Gulf of St. Lawrence | Open.canada | Marine Biology |
| Canadian Health Measures Survey (CHMS) Human Biomonitoring Data for Environmental Chemicals | Open.canada | Environment |
| Results from the 2023 Staffing and Non-Partisanship Survey | Open.canada | Demographics |

# C  EXPERIMENTAL SETUP

## C.1  IMPLEMENTATION DETAILS

All open-source models are sourced from the HuggingFace's `transformers` library (Wolf et al., 2020), and experiments were conducted using $2 \times 48$ GB NVIDIA L40S GPUs. Table 9 lists the API names of closed-source models and the HuggingFace model names of open-source models.

Table 9: List of LLM model names in the experiments

| Model Name | API name or Huggingface model name |
| --- | --- |
| GPT-4o | `gpt-4o-2024-08-06` |
| GPT-4o-mini | `gpt-4o-mini-2024-07-18` |
| Gemini 2.5 Flash | `gemini-2.5-flash` |
| Gemini 2.5 Pro | `gemini-2.5-pro` |
| Devstral-Small | `mistralai/Devstral-Small-2507` |
| Qwen3-30B | `Qwen/Qwen3-30B-A3B-Instruct-2507` |
| Qwen3-Coder-30B | `Qwen/Qwen3-Coder-30B-A3B-Instruct` |
| DeepSeek-R1-14B | `deepseek-ai/DeepSeek-R1-Distill-Qwen-14B` |
| Llama3.1-8B | `meta-llama/Llama-3.1-8B-Instruct` |
| TableGPT2-7B | `tablegpt/TableGPT2-7B` |

Our proposed Insight Agent was configured to generate three initial high-level questions and perform four iterations of its question-answer-insight cycle, resulting in 12 insights. To ensure a fair comparison, the AgentPoirot baseline was configured with parameters that also yielded 12 insights. Furthermore, the summarizing LLM in Insight Agent is instructed to generate the same number of tokens as the summarized sentences from AgentPoirot for a fair comparison.

## C.2  INSIGHT AGENT

The Insight Agent operates through an iterative cycle: it generates questions, answers them using the proposed Answer Agent, and then synthesizes insights from the resulting QA pairs as shown in Figure 4. The insights generated in one step are then used to inform the question generation in the next, creating a continuous exploratory process. This process is governed by a Directed Acyclic Graph (DAG) structure, as illustrated in Figure 9 (b). The graph consists of alternating layers of Question-Answer (QA) nodes and Insight nodes. A new Insight node is generated by synthesizing information from one or more preceding QA nodes, and conversely, a new QA node is generated by drawing upon one or more preceding Insight nodes. Crucially, a new node can be connected to parent nodes from any previous iteration, not just the immediately preceding one. This DAG structure facilitates the aggregation of multiple lines of inquiry, enabling the generation of more diversified and comprehensive insights compared to a simpler tree-based exploration as in Figure 9 (a), where insights from different depths and branches are not connected. The decision of which nodes to aggregate is determined by the reasoning capabilities of LLM; to guide this process, our prompt explicitly instructs the model to consider synthesizing information from multiple parent nodes when possible.

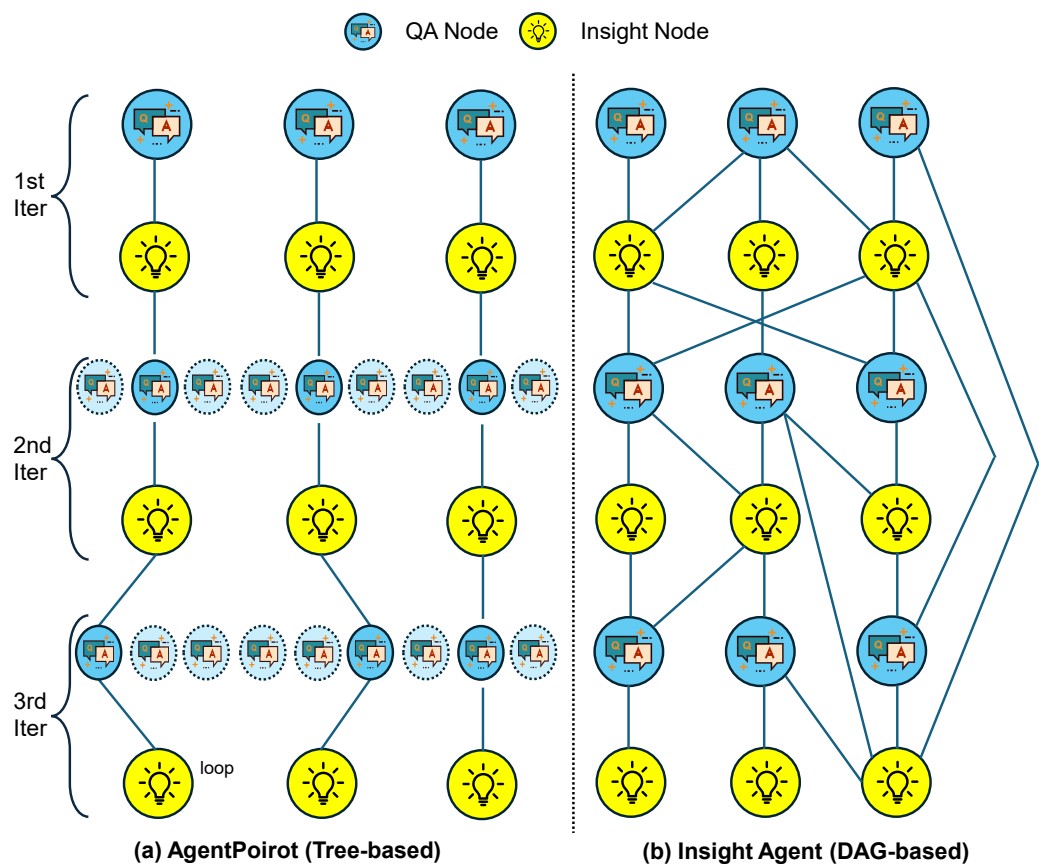

Figure 9: Insight generaion process by (a) tree-based process used in AgentPoirot, where each question is selected by LLM from the question candidates and (b) directed acyclic graph-based process used in Insight Agent

# D RESULTS

## D.1 DETAILS OF TABLE INSIGHT RESULTS

Table 10 shows the dataset-level comparison between AgentPoirot and Insight Agent in terms of the insight-level and summary-level score, and Table 13 presents the notable apple-to-apple comparison among target insight, the most relevant insight from Insight Agent, and one from AgentPoirot, showing the score respectively.

## D.2 COMPUTATIONAL COSTS

Table 11 provides the computational aspects of Answer Agent and Insight Agent when the LLM is Gemini 2.5 Flash and GPT-4o. The table shows an inference latency, the number of input tokens, the number of output tokens, and estimated API cost per execution of both agents.

## D.3 ABLATION STUDIES

**Generalizability of Insight Agent**: To verify that the proposed Insight Agent is unbiased, we evaluated it on existing table insight generation benchmarks using InsightBench (Sahu et al., 2025). Both agents use Gemini 2.5 Flash as the LLM, and we adopt the same evaluation metrics as in the main experiments, namely G-Eval based on GPT-4o. As shown in Table 12, the Insight Agent outperforms AgentPoirot on both insight-level and summary-level scores.

| Dataset | Insight-level | | Summary-level | |
|---|---|---|---|---|
| | AgentPoirot | Insight Agent | AgentPoirot | Insight Agent |
| Cross-Canada Survey of Radon Concentrations in Homes | 0.331 | **0.470** | 0.424 | **0.564** |
| Boston Buildings Inventory | 0.235 | **0.296** | 0.386 | **0.452** |
| Fixed gear sentinel fisheries program - northern Gulf of St. Lawrence | 0.138 | **0.174** | 0.175 | **0.219** |
| Number of Weight Loss Surgeries Performed in California Hospital | 0.332 | **0.368** | **0.457** | 0.442 |
| Canadian Health Measures Survey (CHMS) Human Biomonitoring Data for Environmental Chemicals | 0.347 | **0.363** | **0.481** | 0.462 |
| Results from the 2023 Staffing and Non-Partisanship Survey | **0.315** | 0.220 | 0.232 | **0.290** |
| #Winners | 1 | **5** | 2 | **4** |

Table 10: Dataset-level Table Insight score comparison.

Table 11: Computational Costs of Answer Agent and Insight Agent

| LLM | Answer Agent | | | | Insight Agent | | | |
|---|---|---|---|---|---|---|---|---|
| | Inference time [s] | #Input tokens | #Output tokens | Cost [$] | Inference time [s] | #Input tokens | #Output tokens | Cost [$] |
| Gemini 2.5 Flash | 21.21 | 9.61K | 428 | 0.004 | 278.39 | 261K | 21K | 0.13 |
| GPT-4o | 28.24 | 9.72K | 561 | 0.030 | 334.38 | 243K | 11K | 0.72 |

# E  PROMPTS

In this section, we show the prompts for each stage or module in the benchmark construction, Answer Agent and Insight Agent.

Prompt1: Prompt for question generation in benchmark construction

```
You are a top question generator for the tabular data. Given single or
    multiple tabular data, your task is to generate the high-quality
    insightful question. The question must be answered by using the given
     single or multiple tabular data. Please follow the tips below.

- The question is utilized to retrieve corresponding tabular data from
    data lake, so the question should be de-contextualized enough to
    search the suitable table among tons of data. Data lake has a variety
     of datasets covering a lot of years and locations, so please try to
    specify the specific years and location in the question if available
    by referring to data origin, data title, data description, whatever.
- Do not include ID-like words and codes in the question, and do not
    contain dataset name in the question because human usually do not
    know the dataset names.
- The question should be the one human tends to ask naturally when they
    are interested in the tabular data to get insights deepening human
    understanding.
- Specify answer format and include it in the question. For example, if
    the question asks top 5 areas of something, the format should be the
    list. If the questions asks how does one compare to the other, the
    percentage would be the ideal format.
```

Table 12: Evaluation on InsightBench

| Benchmark | AgentPoirot | | Insight Agent | |
|---|---|---|---|---|
| | Insight | Summary | Insight | Summary |
| InsightBench | 0.3269 | 0.3197 | **0.3275** | **0.3565** |

```
- If you are given multiple tabular data, please generate the question
    involving as many tables as possible.
- If the question type involves multi-turn QA, please list the questions
    in the order of asking in the output format by linking via '/', such
    as 'What is --- ? / What is --- ? / What is --- ? ...'.
- The output format is JSON format where key is question like {
    output_format} without code syntax block, and keys and values should
    be enclosed with double quotes. Please follow the format and do not
    add redundant explanation. You should not include line breaks in JSON
     format.

The type of the question is '{question_key}', and the description of the
    question type is '{question_description}'. Following is the
    information of tabular data.

Data Origin: {data_source}
Data publisher: {publisher}
Data source: {dataset_title}
Data source description: {dataset_description}

Following is the information for tables.
{serialized_table_information}

Following is the external knowledge on the tabular information. Please
    use the knowledge to make the ID-like or ambiguous words clear.
{extenral_knowledge}
```

Prompt2: Prompt for question scoring in the benchmark construction

```
Your task is to select and rank the questions generated from tabular data
    . Given tabular data and list of questions with indices, you must
    select the best 5 questions and answer the list of indices of these
    questions in the ranking order.
You are required to select and rank based on the following perspectives:

- Relevance to the dataset: Directly grounded in the corresponding data
- Actionability and insightfulness: Lead to concrete insights or
    decisions, not just descriptive stats
- Complexity and depth: Require multiple columns or more nuanced
    reasoning
- Clarity and specificity: Clear and unambiguous. Be mindful about the
    output format. If there are possible various output formats based on
    the question, the question is very bad.
- Novelty or non-obviousness: Unexpected relationships or challenging
    assumptions
- Naturalness: The question should be natural for humans, and it is not
    appropriate if the question has a tone nobody asks in that way.
- De-contextualized: The question must be clear enough to retrieve the
    target data from among tons of data, so it should include specific
    datetime and location. This is the highest priority for the question,
     the question without this point is bad.
- No ID like words: The question must not include ID-like words and codes
     (e.g. Region ID 5, Device ID 1229).
- No quoting: The question should avoid quoting column names or cell
    values verbatim (e.g., avoid phrases like indicator named '...' or
    value for category named '...')
```

```
Following is the list of questions.
{questions}

Following is the information of tabular data.
Data Origin: {data_source}
Data publisher: {publisher}
Data source: {dataset_title}
Data source description: {dataset_description}

Following is the information for tables.
{serialized_table_information}

Following is the external knowledge on the tabular information. Please
    use the knowledge to make the ID-like or ambiguous words clear
{external_knowledge}

The output format is JSON format where keys are question and code like {
    output_format} without code syntax block. You should not include line
     breaks in JSON format. Please follow the format and do not add
    redundant explanation. You should not include line breaks in JSON
    format.
```

Prompt3: Prompt for answer genertion in the benchmark construction

```
You are a top data analysis for the tabular data. Given a tabular data
    and question related to the table, your task is to generate the
    Python code to answer the question with the use of methods in Pandas.
     The question must be answered by using the given tabular data.

- If the answer aims to generate the image files (e.g. chart, graph, or
    figure), please set the output file name as 'output_[index].png',
    where 'index' is the index of single or multiple files, and please do
     not include output file names except for in the savefig function.
    Also, please follow the tips below. Create a clear and easy-to-read
    graph for human when the task is visualization, and try to use legend
     as long as the number of labels is not large. When you call 'savefig
    ' function, you must set the following parameter: "bbox_inches='tight
    '".
- The output format is JSON format where keys is code like {output_format
    } without code syntax block when generating Python code, and keys and
     values should be enclosed with double quotes. Please follow the
    format and do not add redundant explanation. You should not include
    line breaks in JSON format. In the Python code, the dataframe is
    tentatively read from 'data_[index].csv', where 'index' is the index
    of the table starting from 1. Also, please include print statement
    for the final answer in the last line.

Question: {question}

Data Origin: {data_source}
Data publisher: {publisher}
Data source: {dataset_title}
Data source description: {dataset_description}
Dataset title: {file_title}
Dataset description: {file_description}

{serialized_table_information}

Following is the external knowledge on the tabular information. Please
    use the knowledge to make the ID-like or ambiguous words clear.
{extenral_knowledge}
```

Prompt4: Prompt for coding module in Answer Agent

```
You are a top data analysis for the tabular data. Given a tabular data
    and question related to the table, your task is to generate the
    Python code to answer the question with the use of methods in Pandas.
     The question must be answered by using the given tabular data.

- If the answer aims to generate the image files (e.g. chart, graph, or
    figure), please set the output file name as 'output_[index].png',
    where 'index' is the index of single or multiple files, and please do
     not include output file names except for in the savefig function.
    Also, please follow the tips below. Create a clear and easy-to-read
    graph for human when the task is visualization, and try to use legend
     as long as the number of labels is not large. When you call 'savefig
    ' function, you must set the following parameter: "bbox_inches='tight
    '".
- If the question includes the specified output format, please follow the
     format without adding redundant texts. Please include the answer in
    the print statement in the last line, and do not use the print
    statement except for the last line. Please do not use print statement
     if the answer aims to generate the image files.
- Please do not truncate the decimal point unless the question requires
    it.
- The output is only Python code without additional explanation.
- In the generated Python code, the input file path is 'data_[index].csv
    ', where 'index' is the index of the table starting from 1 (1, 2, 3,
    4...) depending on the number of available datasets, so for instance
    the code includes 'pd.read_csv('data_1.csv')' when loading the csv
    file. The tentative path will be replaced by the actual path
    afterwards, so please just follow the rule.
- Please include print statement for the final answer in the last line.
- Please do not include try-except statement and exit().

Question: {question}

Data publisher: {publisher}
Data source: {dataset_title}
Data source description: {dataset_description}

This question is a part of multi-turn conversation. Following is the
    previous question and answer pairs. You can refer to them to
    understand the contexts.
{QA Pairs}

{serialized_table_information}

Following is the external knowledge on the tabular information. Please
    use the knowledge to make the ID-like or ambiguous words clear.
{external_knowledge}
```

Prompt5: Prompt for self-correction module in Answer Agent

```
You are a top data analyst with much experience on Python. Given the
    Python code and error message generated by executing the code, your
    task is to revise the Python code. Error message is '{error}', and
    the here is the Python code. Please do not include try-except
    statement. You should output only the code without any other text and
     markdown formatting:

{generated_code}
```

Prompt6: Prompt for visualization reflection module in Answer Agent

```
You are a top data analyst with much experience on Python and matplotlib.
     Given one generated figure that aims to answer the question '{
    question}', your task is to analyze the figures and judge whether the
```

```
      figures are aligning with the question and human understandable. If
   it is not, please revise the Python code to generate figure. Please
   do not include try-except statement. If it is OK, please answer only
   "OK!" without additional text.
Here are the perspectives to consider to judge whether the figure is
   human understandable or not:

- The figure should be clear without too many data points.
- There should not be overlapped colors in the figure.
- Axis labels should be clear and not too long.
- If the figure is a line chart, the line goes to the right direction
   without going back and forth.
- If the figure has small number of data points, the figure should be the
    scatter plot.

Here is the code to generate the figure. If you output the revised code,
   please only output the code without any other text and markdown
   formatting:

{code}
```

Prompt7: Prompt for text-answer reflection module in Answer Agent

```
You are a top data analysis for the tabular data. Given the question
   asking about table data, the generated code to answer the question,
   and the answer, please revise the code toward the correct answer if
   the answer would be wrong. Table information is also given to
   contextualize. If the given answer is correct and the code does not
   need to be revised, please answer only 'OK!' without additional text.
    If the code needs to be revised, please answer only Python code
   without additional explanation and any markdown formatting. Following
    perspectives are included in the correct answer or codes.

- Given questions include particular output formats. Following the output
    format is imperative without adding redundant texts.
- Answers are included in the print statement.
- Please do not include try-except statement.
- The question is a part of the multi-turn questions, the generated code
   should include context produced from the previous QAs if needed.
- Columns in the table data would include the multiple hierarchical
   categories (e.g. total, male, female in gender column). Please be
   careful about the aggregation of the column if needed.
- Columns in the table data would represent numerical values as string
   values (e.g. comma is inserted). Please convert them into numerical
   values appropriately if needed.
- Columns in the table data would include special characters as a
   replacement of NaN values (e.g. x, -, -99999, etc). Please replace
   them with NaN values if needed.
- If the answer is NaN or None, the filtering conditions might be wrong.
- If the output format is just numerical values, they should not be
   truncated or rounded.

## QA pair for the question
Target question: {question}
Generated code that would be revised:
{code}
Answer: {answer}

## QA pairs so far in a multi-turn conversation
{QA_pairs}

## Table information
Data Origin: {data_source}
Data publisher: {publisher}
Data source: {dataset_title}
```

```
Data source description: {dataset_description}

{serialized_table_information}

Following is the external knowledge on the tabular information. Please
    use the knowledge to make the ID-like or ambiguous words clear.
{external_knowledge}
```

Prompt8: Prompt for question generator in Insight Agent

```
You are a top data analyst for the tabular data. Your final goal is to
    generate insights from the table. Insights should be led from the
    informative intermediate results (e.g. graph of data distribution,
    statistical values of certain columns, ranking, aggregated values).
    Therefore, you are required to generate a sequence of good questions
    invoking insightful observations. After answering these questions,
    the insights will be generated based on the intermediate QA pairs.
    Following is the rules to generate questions.

- The number of questions is {number_of_initial_questions}.
- Each question is expected to consider the contexts produced by the
    previous questions.
- Format of questions are connected via vertical lines without adding
    redundant explanation before and after the question parts,
    specifically 'text1 / text2 / text3 / ...'.
- If you refer to column names and cell values in the table, you should
    avoid use exact names by rephrasing them naturally without enclosing
    single/double quotes.
- Questions ask to provide not only text-based simple answer but also
    text-based complicated statistical information (e.g. correlation) and
     visualization (e.g. graph of data distribution, heatmap
    relationships among columns).
- Questions should specify the output format for each question, saying
    that 'Please provide as a line chart', 'Please answer as a numerical
    value', etc...

Following is table information.
## Table Information
Data Origin: {data_source}
Data publisher: {publisher}
Data source: {dataset_title}
Data source description: {dataset_description}

{serialized_table_information}

Following is the external knowledge on the tabular information. Please
    use the knowledge to make the ID-like or ambiguous words clear.
{external_knowledge}
```

Prompt9: Prompt for follow-up question generator in Insight Agent

```
You are a top data analyst for the tabular data. Your final goal is to
    generate insights from the table. Insights should be led from the
    informative intermediate results (e.g. graph of data distribution,
    statistical values of certain columns). Therefore, you are required
    to generate a sequence of good follow-up questions invoking
    insightful observations by referring to questions and insights
    generated in the previous steps. After answering these questions, the
     insights will be generated based on the intermediate QA pairs.
    Following is the rules to generate questions.

- The number of questions is {number_of_questions}.
- Follow-up questions are generated by referring to the single or
    multiple insights, and you must include which insights are referred
```

```
    to in the output. Therefore, output format is {output_format} in the
    strict JSON format without markdown formatting and indentation.
    Please do not add additional explanation.
- It is encouraged to aggregate multiple insights to generate single
    follow-up question.
- Each insight composes of not only insight text but also the question
    numbers that the insight is generated from, so please also refer to
    these questions.
- Avoid duplication of questions by referring to the generated questions
    so far.
- If you refer to column names and cell values in the table, you should
    avoid use exact names by rephrasing them naturally without enclosing
    single/double quotes.
- Questions must include mixing text-based simple answer, text-based
    complicated statistical information (e.g. correlation) and
    visualization (e.g. graph of data distribution, heatmap relationships
     among columns).
- Questions should specify the output format for each question, saying
    that 'Please provide as a line chart', 'Please answer as a numerical
    value', etc...

## Questions so far
{previous_questions}

## Insights so far
{previous_insights}

Following is table information.
## Table Information
Data Origin: {data_source}
Data publisher: {publisher}
Data source: {dataset_title}
Data source description: {dataset_description}

{serialized_table_information}

Following is the external knowledge on the tabular information. Please
    use the knowledge to make the ID-like or ambiguous words clear.
{external_knowledge}
```

Prompt10: Prompt for insight generator in Insight Agent

```
You are a top data analyst for the tabular data. Your task is to generate
    insights that can be read from table information QA pairs related to
    the table. Insights should follow the following points.

- Insights are text-based, and should include factual and informative
    information that attract readers. Also, they are expected to invoke
    non-trivial realizations for humans.
- Insight should be generated from QA pairs, and please include which QA
    pairs contribute to the insight by referring to the corresponding
    question numbers.
- Write down {number_of_insights} insights, and the output format is {
    output_format} in the strict JSON format without markdown formatting }
    and indentation. Do not add additional texts or redundant information
    .
- Single insight can be produced from one or multiple QA pairs. It is
    encouraged to aggregate multiple QA pairs to generate single insight.
- Avoid logical leap under many uncertain assumptions.
- Avoid duplications of insights by referring to the insights generated
    so far.

Following is QA pairs and table information.

## QA Pairs so far
```

```
{previous QA pairs}

## Insights so far
{previous_insights}

## Table Information
Data Origin: {data_source}
Data publisher: {publisher}
Data source: {dataset_title}
Data source description: {dataset_description}

{serialized_table_information}

Following is the external knowledge on the tabular information. Please
    use the knowledge to make the ID-like or ambiguous words clear.
{external_knowledge}
```

Prompt11: Prompt for fine-grained analysis of Table Insight

```
You are an expert data analyst. Your task is to evaluate a 'Predicted
    Insight' against a 'Target Insight' by referring to the score
    measuring how close the predicted insight is close to target insight.
     You must assess the prediction based on the four perspectives
    defined below. For each perspective, you must answer only with the
    score (1 to 5, 1 is the lowest and 5 is the highest).

Definitions:
Topic Relevance: Does the prediction address the same topic as the target
    ?
Quantitative Details Match: Does the prediction mention the same specific
     quantitative numbers as the target?
Qualitative Details Match: Does the prediction mention the same specific
    names (e.g., places, trends, proper names) as the target?
Narrative Alignment: Does the prediction make the same core argument or "
    headline" conclusion as the target?

Target Insight:
{gt_insight}

Predicted Insight:
{pred_insight}
score: {score}

The output format is {output_format} in the valid JSON format without any
     additional text.
```

Prompt12: Prompt for MLLM-as-a-judge

```
You are a top data analyst. Given the generated graph(s) (first {num_vis}
     images), the target graph(s) (last {num_tgt} images), and Python
    codes to generate them, your task is to judge whether the generated
    graph is the ROUGHLY same as the target graph based on the following
    perspectives attached with reasons of judgement. Output format is {
    output_format} in JSON format enclosed with double quotation.

- Chart type (e.g. Line chart, bar chart, pi chart...)
- Roughly similar axis name / legends (They should not be exactly the
    same, but rough similarity is preferred)
- Values of the characteristic data points should be matched
- IMPORTANT: Prioritize the trajectory of the graph because the graphs
    tend to be matched if the shapes of the graphs are the same
- Ignore the color scheme and the size of the chart
- Ignore the position (vertical or horizontal) of the subplots in the
    graph
```

```
- Ignore the marker in the line chart
- Ignore the grid of the chart
- Ignore the title of the graph
- Ignore the rotation of the axis
- If the x-axis is year or date, as long as the minimum and maximum date
    is the same, it is regarded as the same in terms of the x-axis
- There is a case where tick marks are not explicitly drawn though the
    range of the axis is the same. In that case, please treat them as the
     same graph

Please answer "Same" or "Different". If the generated graph matched the
    target graph based on the above perspectives, please say "Same".
    Otherwise, please answer "Different". Please do not add explanation.
    Following is the Python codes for generated graph and target graph.

### Code for generated graph
{pred_code}

### Code for target graph
{target_code}
```

Prompt13: Prompt for Insight-leve/Summary-level Score

```
You are a high school teacher evaluating student responses to a question.
     You are tasked with grading the response based on how well it
    answers the question. You are to provide a numerical rating for how
    well the provided response matches the ground truth answer.

Below is an instruction that describes a task. Write a response that
    appropriately completes the request.

### Instruction:
Provided Answer:
{answer}

Ground Truth Answer:
{gt_answer}

Follow these instructions when writing your response:
* On a scale of 1-10, provide a numerical rating for how close the
    provided answer is to the ground truth answer, with 10 denoting that
    the provided answer is the same as ground truth answer.
* Your response should contain only the numerical rating. DONOT include
    anything else like the provided answer, the ground truth answer, or
    an explanation of your rating scale in your response.
* Wrap your numerical rating inside <rating></rating> tags.
* Check very carefully before answering.
* Follow the output format as shown in the example below:
Example response:
<rating>7</rating>

### Response:
```

| GT | Insight Agent | AgentPoirot |
|---|---|---|
| Provinces/Territories with Highest Prevalence: The provinces and territories that exhibited the highest percentages of participant homes testing above the radon guideline were **Manitoba, New Brunswick**, Saskatchewan, and the **Yukon**. For example, **New Brunswick had a raw percentage of 24.8%** and a population-weighted percentage of 20.6% of homes above 200 Bq/m³, while Manitoba had 23.7% raw and 19.4% population-weighted. | Radon concentrations vary significantly across Canadian provinces and territories, with **New Brunswick, Yukon, and Manitoba** consistently showing the highest average concentrations and a wider spread of values. **Notably, New Brunswick has the highest proportion of homes exceeding the 200 Bq/m³ mitigation guideline, with over 25% of its homes above this threshold**. Within these high-concentration regions, specific health regions and Forward Sortation Areas (FSAs) exhibit even higher localized averages, highlighting the importance of granular regional analysis for targeted mitigation efforts. (score: 0.78) | The average radon concentration in **New Brunswick** (179.9 Bq/m3) is more than double that of British Columbia (70.98 Bq/m3) and the Northwest Territories (70.96 Bq/m3), highlighting significant regional disparities in radon levels across Canada. (score: 0.37) |
| Localized Risk in Provinces with Lower Averages: **Even in provinces where the overall population-weighted results indicated a lower incidence of homes with elevated radon levels, there were still specific Health Regions with high radon levels**. For example, in **Ontario**, where the population-weighted estimate was 4.6% of homes exceeding the guideline, 13 of 36 Health Regions (over one-third) had more than 10% of homes test above the guideline. | While New Brunswick and Yukon have the highest average radon concentrations, Ontario and Manitoba also show significant radon concerns, particularly regarding high outliers. **Ontario** has a substantial number of measurements exceeding 500 Bq/m³ (49 instances), and Manitoba has a high proportion of measurements above 200 Bq/m³ (23.67%), second only to New Brunswick. **This indicates that even provinces with lower overall average radon concentrations can have localized areas with very high radon levels, necessitating targeted mitigation efforts.** (score: 0.64) | The significant variability in radon concentrations between Health Regions and their provincial averages, as highlighted by the large standard deviation of 40.21 Bq/m3, suggests that localized geological factors or housing characteristics within specific Health Regions may play a more dominant role in radon levels than broader provincial trends. (score: 0.49) |
| Age-Related Increases in Mirex and Marker PCBs: Mirex concentrations increased with age in Cycle 1 (2007–2009), with the highest mean serum concentration found in the 60–79 years age group (0.019 ng/g serum) compared to younger groups (e.g., 0.0014 ng/g serum for 6–11 years). **Similarly, the sum of Marker PCBs (PCB 138, 153, 180) generally showed an increase in concentrations with increasing age across all cycles**, with the 60–79 years age group consistently exhibiting the highest arithmetic means (e.g., 140 ng/g lipid in Cycle 1). | **Polychlorinated biphenyls (PCBs), particularly 'Marker polychlorinated biphenyls (sum of PCB 138, 153, 180)', show a clear age-related accumulation**, with significantly higher average measured amounts in older age groups (40-79 years) compared to younger ones (3-39 years). This suggests a persistent presence and bioaccumulation of these substances over a person's lifetime. (score: 0.71) | For Polychlorinated biphenyls, the AM-MA values for the 'Total' gender group consistently increase with age, with the 60-79 age group showing AM-MA values as high as 9.62, significantly higher than the 12-19 age group which has values as low as 0.89. (score: 0.31) |

Table 13: Apple-to-apple comparison among GT insight, insight from Table Insight, and one from AgentPoirot.

