# OpenReview forum: "OpenDataBench: Real-World Benchmark for Table Insight Generation and Question Answering Over Open Data"
_ICLR.cc/2026/Conference — Submitted to ICLR 2026_

### Official Review · Reviewer_SPHH · 2025-10-27

**Soundness:** 3
**Presentation:** 3
**Contribution:** 2
**Rating:** 2
**Confidence:** 4

**Summary:**

The paper focuses on the challenge of evaluating LLMs on real-world data analysis, where existing table reasoning benchmarks fail to capture the complexity of large, multi-table datasets and the open-ended nature of insight generation.
To address this, the authors construct OpenDataBench, a benchmark built from governmental open data featuring tasks for multifaceted table QA and Table Insight Generation, and introduce two corresponding agentic solutions, including an Answer Agent and an Insight Agent. Experiments show that even state-of-the-art models like GPT-4o and Gemini 2.5 perform poorly on these tasks, revealing a significant gap between current LLM capabilities and realistic data-analysis demands.

**Strengths:**

1. The paper targets a real and important challenge that current LLM benchmarks for table reasoning do not reflect the complexity and messiness of real-world data analysis.

2. The idea of building a realistic benchmark from government open data is solid and practical, offering a credible way to capture large-scale, multi-table, and heterogeneous data scenarios often encountered in real applications.

**Weaknesses:**

1. The novelty is limited. The benchmark mainly extends existing table QA and insight-generation setups to larger, real-world datasets without introducing fundamentally new tasks or evaluation methods.

2. The proposed agents (Answer Agent and Insight Agent) are largely incremental combinations of existing techniques like code generation, self-correction, and reflection, offering engineering value but little conceptual innovation.

3. The evaluation results mainly confirm that current LLMs struggle with large and messy tables, rather than revealing new insights into how to overcome these challenges.

**Questions:**

see weakness

**Details Of Ethics Concerns:**

no ethics concerns

---

> ### Author Response · Authors · 2025-11-25
> **Author Response to Reviewer SPHH**
>
> We thank the reviewer for sharing the critical questions about our benchmark. We would like to answer each question as below.
>
> > The novelty is limited. The benchmark mainly extends existing table QA and insight-generation setups to larger, real-world datasets without introducing fundamentally new tasks or evaluation methods.
>
> We thank the reviewer for this thoughtful comment. We would like to emphasize that OpenDataBench differs from existing benchmarks in a fundamental way: it focuses on large, heterogeneous, and multi-table datasets that exceed the LLM context window, a scenario not covered by prior benchmarks. For instance, we have observed that Gemini 2.5 Flash achieves strong performance on TableBench (0.74 on fact-checking and 0.65 on numerical reasoning), largely because the small tables in TableBench can be fully serialized within the prompt. In contrast, OpenDataBench contains complex multi-tabular datasets that require agentic strategies and specialized table handling to produce answers—behaviors that existing benchmarks do not meaningfully evaluate. While we do not introduce new task definitions or evaluation protocols, we believe the shift to real-world and large-scale data constitutes a significant contribution by exposing limitations of current LLMs that have not been measured previously.
>
> > The proposed agents (Answer Agent and Insight Agent) are largely incremental combinations of existing techniques like code generation, self-correction, and reflection, offering engineering value but little conceptual innovation.
>
> We appreciate the reviewer’s observation and would like to clarify two conceptual contributions of both Answer Agent and Insight Agent.
>
> - Feature-type–aware table serialization.
>   - We introduce a tailored serialization scheme that encodes table columns according to their feature types to maximize the amount of table information fitting into the prompt as shown in Appendix B.2.2. Our ablation study (Section 4.3) shows that this method improves performance of Answer Agent by approximately 16% with Gemini 2.5 Flash, demonstrating that this is not merely an engineering refinement but an essential methodological component for both agents.
> - DAG-based insight exploration.
>   - Our Insight Agent builds a directed acyclic graph to aggregate context across all previously generated questions, answers, and insights, enabling multi-perspective and layered insight construction as explained in Appendix C.2 in detail. Prior works (e.g. AgentPoirot) typically rely only on single-step or sequential reasoning, whereas our DAG-based approach supports richer, more compound forms of insight discovery.
>
> > The evaluation results mainly confirm that current LLMs struggle with large and messy tables, rather than revealing new insights into how to overcome these challenges.
>
> We thank the reviewer for these helpful comments. While our quantitative evaluation indeed highlights the limitations of current LLMs, our qualitative analysis provides concrete guidance toward addressing these challenges as noted in Section 4.2. Specifically, for Table QA, the most common failures arise from Condition Filter Error and Data Transformation Error, suggesting that more compact yet informative table representations (e.g., EDA-based summaries) could improve reasoning accuracy. For Table Insight, Figure 6 shows that models frequently miss accurate quantitative values, implying that improved QA accuracy is a prerequisite for producing reliable insights. In the revised version, we will emphasize how these findings illuminate actionable directions for designing more capable LLM-based systems for table-based real-world deployment in the section about qualitative analysis.

---

### Official Review · Reviewer_n5Wt · 2025-10-28

**Soundness:** 2
**Presentation:** 2
**Contribution:** 2
**Rating:** 4
**Confidence:** 2

**Summary:**

This paper introduces OpenDataBench, a benchmark designed to evaluate table understanding capabilities of language models using real-world open data from governmental sources. The benchmark addresses two limitations of existing table reasoning benchmarks: lack of real-world complexity (large-scale, multi-table datasets with external knowledge) and narrow task scope (focusing only on question answering while neglecting insight generation). OpenDataBench features two main tasks: Table QA (answering complex decomposable questions with text or visual outputs) and Table Insight (generating expert-level findings from exploratory analysis). The authors also propose two agentic solutions: an Answer Agent with fail-safe modules for Table QA and an Insight Agent using graph-based exploration for Table Insight.

**Strengths:**

1 The proposed dataset represents a good advancement over existing benchmarks that use small, clean tables.

2 The benchmark formalizes both reactive (Table QA) and proactive (Table Insight) data analysis tasks.

3 The paper employs a four-stage pipeline (question generation, scoring, answer generation, human verification) with multiple LLM judges and human experts.

**Weaknesses:**

1 The use of LLM-based evaluation for insight generation, while practical, introduces subjectivity. The quantitative metrics may not fully capture insight quality, but provides limited validation of the evaluation methodology's reliability.

2  The multi-agent approaches involve complex pipelines with multiple LLM calls, but the paper lacks analysis of computational requirements, inference latency, or cost considerations, which are important factors for real-world deployment.

**Questions:**

please refer to the weakness

---

> ### Author Response · Authors · 2025-11-25
> **Author Response to Reviewer n5Wt**
>
> Thank you for noting “a good advancement” for our benchmark, and providing critical questions. We would like to address each question as below.
>
> > The use of LLM-based evaluation for insight generation, while practical, introduces subjectivity. The quantitative metrics may not fully capture insight quality, but provides limited validation of the evaluation methodology's reliability.
>
> Thank you for raising this important point regarding potential subjectivity in LLM-based evaluation. To assess the reliability of our evaluation protocol, we conducted an analysis to measure how well the proposed insight-level scores align with human judgments.
>
> Specifically, we sampled 50 pairs of generated insights (Insight A and Insight B) for each ground-truth insight and asked three independent annotators to assess which of the two was closer to the ground truth. Annotators selected one of five relative options: A+ (A is definitely better), A (A is slightly better), N (comparable), B (B is slightly better), or B+ (B is definitely better). Each option was mapped to a normalized score in {-2, −1, 0, 1, 2}. We averaged the three annotators’ scores to obtain the final human-judgment score for each pair.
>
>
> In parallel, we computed a normalized score from the insight-level metric based on the difference in metric scores (score for B minus score for A). We then evaluated the agreement between human judgments and the metric-derived scores using both Pearson and Spearman correlations. The resulting correlations were 0.669 and 0.663, respectively. These results demonstrate a substantial alignment between the proposed metric and human perception, providing empirical support for the reliability of our evaluation methodology.
>
> We will add the evaluation in Section 4.1.
>
> > The multi-agent approaches involve complex pipelines with multiple LLM calls, but the paper lacks analysis of computational requirements, inference latency, or cost considerations, which are important factors for real-world deployment.
>
> Thank you for suggesting this practical point. We agree that understanding computational requirements and cost is essential for assessing the practicality of the LLM-based system in the real-world deployment. To address this, we have added an analysis of inference latency, token usage, and estimated API cost for each execution of both Answer Agent and Insight Agent as shown in the table below. We will incorporate the result in the revised paper to show the resource requirements of our approach in Appendix D.2.
>
> | LLM              | Agent         | Inference time [s] | #input tokens | #output tokens | Cost [$] |
> | ---------------- | ------------- | ------------------ | ------------- | -------------- | -------- |
> | Gemini 2.5 Flash | Answer Agent  | 21.21              | 9.6K          | 428            | 0.004    |
> | Gemini 2.5 Flash | Insight Agent | 278.39             | 261K          | 21K            | 0.13     |
> | GPT-4o           | Answer Agent  | 28.24              | 9.7K          | 561            | 0.030    |
> | GPT-4o           | Insight Agent | 334.38             | 243K          | 11K            | 0.72     |

---

### Official Review · Reviewer_32SQ · 2025-10-31

**Soundness:** 3
**Presentation:** 3
**Contribution:** 2
**Rating:** 4
**Confidence:** 4

**Summary:**

The authors propose a new benchmark, OpenDataBench, which consists on two tasks: multi-faceted table question-answering and table insights. The former assesses factual reasoning over composable questions (with multiple subquestions), while the latter, challenges models to provide expert-level insights , which can require in-depth analysis. The authors leverage a filtering technique, human annotators  and LLM as a judge in the construction of the data for both tasks. The tasks are based on publicly available complex, heterogeneous datasets that contain large tables. The authors propose a table serialization technique to bypass the need to pass the full table to the LLM. The authors also provide two agents for each task which outperform existing models and agents. The authors also include error analysis and an ablation wrt the proposed table serialization.

**Strengths:**

- The authors propose a complex enough benchmark that can be leveraged to assess agentic workflows. This is an active area of research and it is important to have access to realistic benchmarks for evaluation.

**Weaknesses:**

- The paper is missing precise implementation details for the new proposed agents

**Questions:**

- Add one liner to explain how different types of variables are handled in the main paper so there’s no need to go to the appendix to understand the full details.
- It would be good to add more clearly the percentage of discarded data points and questions as well as describe how many times you require to prompt an LLM to provide a better notion of how expensive the dataset creation is.
Even though there is a section that discusses error analysis, one particular failure mode that is not discussed in detail is what happens if there are questions that use multiple tables and the agent fails to fetch all the relevant tables?
- Typically, SQL is the tool of choice to extract information from one or more tables. Is it the case that the python code can leverage libraries to perform SQL queries on the tables?
- It would be interesting to add to the ablation study if instead of naively passing the first 10 rows, what if we pass the table schema and a number of rows? Is that equivalent to the proposed feature serialization?

---

> ### Author Response · Authors · 2025-11-25
> **Author Response to Reviewer 32SQ (Part1)**
>
> We thank the reviewer for providing insightful comments and suggestions for improving the paper quality. We would like to answer each question below.
>
> > Add one liner to explain how different types of variables are handled in the main paper so there’s no need to go to the appendix to understand the full details.
>
> We thank the reviewer for this helpful suggestion. We will add a concise one-sentence explanation in the main text to summarize our handling of variable types, reducing the need to cross-reference the appendix.
>
> > It would be good to add more clearly the percentage of discarded data points and questions as well as describe how many times you require to prompt an LLM to provide a better notion of how expensive the dataset creation is.
>
> Thank you for your valuable advice. We agree that illustrating the proportion of discarded data points and the cost of dataset creation improves the transparency of the pipeline.
>
> Dataset pipeline construction requires 9246 LLM calls, broken down into 736 for the question generation, 736 for scoring, 7360 for the answer generation, and 414 for rephrasing.
>
> After the scoring stage, the pipeline generated 1,840 questions, which were curated down to 211 final questions after removing those that did not meet quality standards. The table below summarizes the proportion of discarded questions by reason. We will add the table as Table 6.
>
> | Reason                                                       | Count (Percentage) |
> | ------------------------------------------------------------ | ------------------ |
> | Unanimous agreement                                          | 114 (6.2 %)        |
> | Dataset lacks sufficient external knowledge to answer        | 550 (29.9 %)       |
> | Question is too ambiguous to construct the definitive answer | 354 (19.2 %)       |
> | Not insightful questions                                     | 611 (33.2 %)       |
>
>
> Distribution of the question types in 211 questions is shown in the below table, which will be introduced in Figure 3 (d).
>
> | Question Type                 | Count (Percentage) |
> | ----------------------------- | ------------------ |
> | Multi-turn Insight Generation | 65 (30.95%)        |
> | Multi-turn Follow-up          | 50 (23.81%)        |
> | Ranking                       | 36 (17.14%)        |
> | Aggregation                   | 21 (10.00%)        |
> | Complex Data Transformation   | 13 (6.19%)         |
> | Counting                      | 12 (5.71%)         |
> | Multi-hop Numerical Reasoning | 11 (5.24%)         |
> | Multi-hop Lookup              | 2 (0.95%)          |
>
> > Even though there is a section that discusses error analysis, one particular failure mode that is not discussed in detail is what happens if there are questions that use multiple tables and the agent fails to fetch all the relevant tables?
>
> This issue corresponds to the error type we refer to as “Wrong choice of tables.” in Figure 5. In some cases, the agent fails to retrieve all relevant tables. When this happens, the final answer is typically incorrect. For instance, if each table corresponds to a different year and the question asks for a trend across time, missing a table leads to a visualization that omits part of the timeline, resulting in an incomplete or incorrect trend. We will polish the example case of the error in the revised version.
>
> > Typically, SQL is the tool of choice to extract information from one or more tables. Is it the case that the python code can leverage libraries to perform SQL queries on the tables?
>
> We thank the reviewer for the interesting suggestion. In the current implementation, both Answer Agent and Insight Agent use Pandas operations exclusively for table manipulation. We chose Python due to the flexible ecosystem to handle various data analysis and data wrangling tasks, such as advanced statistical analysis beyond calculation of basic values, data visualization, and complex datetime handling, which are important to answer the questions in OpenDataBench. We suppose that SQL has difficulties with handling those tasks. However, we understand that SQL works better than Python scripts for the specific tasks (e.g. data merge, data filtering), and the utilization of SQL from Python would be a beneficial future direction. We will incorporate this extension in Section 4.3.1.

---

> ### Author Response · Authors · 2025-11-25
> **Author Response to Reviewer 32SQ (Part 2)**
>
> > It would be interesting to add to the ablation study if instead of naively passing the first 10 rows, what if we pass the table schema and a number of rows? Is that equivalent to the proposed feature serialization?
>
> Thank you for the insightful suggestion. We agree that comparing our proposed serialization with a baseline that includes both schema and a lot of rows is a valuable additional ablation.
>
> Following your suggestion, we conducted an experiment where we provided the table schema (column names, Pandas data types, percentage of missing values, and percentage of unique values) together with a larger subset of rows. For Gemini 2.5 Flash, we supplied the first 200 rows. For GPT-4o, we included  table schema and up to 200 rows but dynamically reduced the number as needed to fit within its 128K context window. The results are summarized in the table below.
>
> We found that this variant performs worse than the naïve baseline that simply passes the first 10 rows. A likely explanation is that LLMs struggle to locate specific keys or values for data manipulation when presented with a large pool of numerical or string entries (most of them might be duplicated). Moreover, 200 rows cover a negligible portion of OpenDataBench tables, whose average size is 212K rows, preventing the LLMs from accessing the large portions of tables.
>
> In contrast, our proposed feature-type (FT)–based serialization explicitly summarizes each column and thus provides a compact yet information-dense representation of the entire table. This appears to support reasoning over large heterogeneous tables and leads to substantially improved performance, as shown below. We will incorporate this additional experiment into the final version to more clearly demonstrate the effectiveness of our serialization approach in Section 4.4.
>
> |                                 | Gemini 2.5 Flash | GPT-4o |
> | ------------------------------- | ---------------- | ------ |
> | First 10 rows                   | 0.310            | 0.242  |
> | First hundreds of rows + Schema | 0.275            | 0.232  |
> | Proposed FT-based serialization | 0.360            | 0.257  |

---

### Official Review · Reviewer_UM5Z · 2025-10-31

**Soundness:** 1
**Presentation:** 3
**Contribution:** 2
**Rating:** 2
**Confidence:** 3

**Summary:**

This paper proposed a new benchmark targeting two tasks -- table question answering over large tables, and table insight generation.
The source tables are collected from public government datasets, and question-answer pairs are generated with LLM followed by human verification, keeping only QA pairs with low execution-result agreement for LLM generated code. Table Insights are extracted from publicly written report based on the tables. Baselines together with error analysis are also provided.
Main contributions of this paper include
1. Created a benchmark for QA over large tables
2. While not the first to create a insights generation benchmark, the insights are extracted from organic sources (human written reports) rather than planted into tables

I think the motivation for this work is clear and if the idea is executed properly this can be an important contribution, but some sections (dataset construction/eval metrics/baseline) lacks sufficient detail to judge soundness of the work, so for now I'm recommending reject.

**Strengths:**

1. This benchmark addresses current limitations/shortcomings in existing table-based QA
2. The insights are collected organically from human written reports, which seems more representative of human interests compared to existing benchmark
3. The answers are not limited to textual format but also covers data visualization/chart generation

**Weaknesses:**

1. Missing some details in dataset construction/eval metrics/baseline design & analysis

    a. Unclear to me how insights are extracted from the human written reports. Also through prompting?

    b. For table QA, is your naive baseline (feeding first 10 rows to LLM in single turn) generating code to produce answer or just prompted to generate output?

    c. baseline analysis -- what's the proportion of answer agent needing to go back to revise the generated code? how many iterations do you allow answer agent to run before stopping

    d. Insight agent -- the pipeline seems similar to the data construction stage for your table QA benchmark generation (except human verification)? how do you determine what questions to keep and what's the stopping criteria? how do you rank the generated questions?

    e. Insight generation metrics -- the proposed G-eval based score compares generated insights against 'ground-truth' insights, but original G-eval score is comparing summary against source article. Also, G-eval reports four separate scores (Coherence, Consistency, Fluency, Relevance). Why was only one combined score reported? How are those different aspects combined to get a single score? I think the description of the proposed g-eval inspired metrics does not have sufficient detail.

    f. Analysis: as I understand, the answer are generated using LLM generated code, and the answer agent is also prompted to generate code, even though the dataset construction employs multiple LLM providers to mitigate bias, is there any chance that the ones that are answered correctly also happens to be generated with the same model? i.e. in the successfully answered portions, say by Gemini, any chance they just happened to be questions whose answers were already generated by Gemini?




2.  Missing some reference
    a. How does this dataset compare to DataBench (Grijalba et al 2024) -- this also seems to be targeting QA over large tables

Jorge Osés Grijalba, L. Alfonso Ureña-López, Eugenio Martínez Cámara, and Jose Camacho-Collados. 2024. Question Answering over Tabular Data with DataBench: A Large-Scale Empirical Evaluation of LLMs. In Proceedings of the 2024 Joint International Conference on Computational Linguistics, Language Resources and Evaluation (LREC-COLING 2024), pages 13471–13488, Torino, Italia. ELRA and ICCL.

**Questions:**

1. Seems like question types are limited to those that are answerable with python code, as the dataset construction process with python execution for answer generation seems to guarantee that. What's the justification behind limiting it to these types of questions? How are you prompting the LLM to generate questions?

2. I appreciate the authors acknowledged the potential subjectivity of insights and their attempt to address it. I'm curious if the authors have conducted analysis on annotator agreement for the generated insights

3. "After executing the code from all four models, we measured the answer consensus to filter out questions that yielded unanimous agreement across all four LLMs. Such instances were deemed to indicate a low level of analytical complexity, making them unsuitable for a benchmark to challenge state-of-the-art models." =>

    a. how do you measure answer consensus?

    b. what is the justification for the claim that 'unanimous agreement' indicates 'low level of analytical complexity' -- seems like this is just to intentionally construct an adversarial set that is hard for current systems, not so much representative of true distribution/human interests?

    c. if the executed code has no agreement, how do you decide which answer to keep?

4. Human verification stage -- "During this stage, annotators also filtered out questions for qualitative reasons,
such as not being insightful, being too ambiguous to permit a definitive answer, or requiring
external knowledge that was unavailable." => what is considered 'insightful'?

---

> ### Author Response · Authors · 2025-11-25
> **Author Response to Reviewer UM5Z (Part 1)**
>
> Thank you for finding the potential significance of the paper and providing a lot of clarification and insightful comments that we need to be clear about. We would like to address each question.
>
> > Unclear to me how insights are extracted from the human written reports. Also through prompting?
>
> We thank the reviewer for this question. The extraction procedure depends on the format of the original reports. Some reports present insights as bullet points; in these cases, we directly treat each bullet point as one insight. Other reports express insights in free text. For these, we upload the reports to NotebookLM [1] and prompt it to suggest ten insights based on the results section. We then manually verify the quality of the extracted insights. We will clarify this process in Section 2.1.3.
>
> [1] Google, NotebookLM, accessed July 10, 2025. URL https://notebooklm.google.com/
>
> > For table QA, is your naive baseline (feeding first 10 rows to LLM in single turn) generating code to produce answer or just prompted to generate output?
>
> The naïve baseline generates the Python code, which is then executed to obtain the final answer. We will elucidate the output form of the baseline in Section 4.2.
>
> > baseline analysis -- what's the proportion of answer agent needing to go back to revise the generated code? how many iterations do you allow answer agent to run before stopping
>
> Thank you for asking for the important clarifications. Answer agent applies two extra steps, self-correction and reflection, to revise code generation when necessary. Maximum of 3 iterations are allowed. Therefore, the revision happens at most six times in each execution. The table below shows the distribution of the revision counts made by Answer Agent based on Gemini 2.5 Flash. As shown, more than 78% of questions require at least one revision, and most of them involve revision once. We will specify the max number of revisions for each component in Section 3.1.
>
> | Number of revisions | Count (Ratio) |
> | ------------------- | ------------- |
> | 0                   | 88 (0.213)    |
> | 1                   | 283 (0.684)   |
> | 2                   | 23 (0.056)    |
> | 3                   | 2 (0.009)     |
> | 4                   | 6 (0.014)     |
> | 5                   | 12 (0.029)    |
> | 6                   | 0 (0.000)     |
>
> > Insight agent -- the pipeline seems similar to the data construction stage for your table QA benchmark generation (except human verification)? how do you determine what questions to keep and what's the stopping criteria? how do you rank the generated questions?
>
> The question and answer generation components in the Insight Agent share similarities with those used in the QA benchmark generation pipeline except that the former does not take the question types listed in B.2.1 as the input.
>
> The question generation step in Insight Agent does not filter or rank the generated questions. It generates 3 best questions in each iteration. The agent runs four iterations of generating questions, answers, and insights, which result in total of 12 questions (3 questions × 4 turns) as noted in C.1.
>
> > Insight generation metrics -- the proposed G-eval based score compares generated insights against 'ground-truth' insights, but original G-eval score is comparing summary against source article. Also, G-eval reports four separate scores (Coherence, Consistency, Fluency, Relevance). Why was only one combined score reported? How are those different aspects combined to get a single score? I think the description of the proposed g-eval inspired metrics does not have sufficient detail.
>
> Thank you for the helpful comment. Our insight-generation evaluation is based on InsightBench, which employs an LLaMA-evaluator—an LLaMA-3–based adaptation of G-Eval that replaces GPT-4. In InsightBench, the evaluator is prompted to output a single rating (1–10) measuring how close the predicted answer is to the ground-truth answer. Therefore, we report this single score as defined by the benchmark. We recognize that referring to the adapted evaluator as “G-Eval” could cause confusion given its differences from the original formulation. To avoid ambiguity, we will revise the terminology and describe the evaluation protocol clearly. We will also include the full prompt for the insight-generation metric in Appendix E (Prompt 13).

---

> ### Author Response · Authors · 2025-11-25
> **Author Response to Reviewer UM5Z (Part 2)**
>
> > Analysis: as I understand, the answer are generated using LLM generated code, and the answer agent is also prompted to generate code, even though the dataset construction employs multiple LLM providers to mitigate bias, is there any chance that the ones that are answered correctly also happens to be generated with the same model? i.e. in the successfully answered portions, say by Gemini, any chance they just happened to be questions whose answers were already generated by Gemini?
>
> Thank you for raising the point that we need to be clear about. In our dataset, no question–answer code pairs generated by an LLM are used directly as the final target pairs. As for the questions, we rephrased the original question by enhancing the naturalness, paraphrasing the column names, and including the specific output formats, so text representations become totally different from the original question. On the other hand, we carefully reviewed the generated codes by LLMs. Among 211 generated codes, 175 questions (82.9%) are revised entirely to match the correct answers, and the remaining 36 questions (18 from GPT-4o, 6 from Gemini 2.0 Flash, 6 from GPT-4o-mini, and 6 from Gemini 2.5 Pro) are modified to match the output format specified by the questions. Therefore, even when an evaluation model (GPT-4o or Gemini 2.5 Flash) answered original questions correctly, it might not mean they have the advantages because both the questions and the corresponding code have been substantially transformed.
>
> > Missing some reference a. How does this dataset compare to DataBench (Grijalba et al 2024)
>
> We thank the reviewer for the helpful suggestion. DataBench is indeed an important benchmark for large tabular datasets, and we acknowledge its relevance to our work. Compared to DataBench, our OpenDataBench substantially extends this line of research by incorporating more complex tasks (e.g., decomposable QA and visualization), multi-table settings, and heterogeneous data sources (including external knowledge). We will include DataBench in Table 1 and add the appropriate citation in the revised manuscript.
>
> > Seems like question types are limited to those that are answerable with python code, as the dataset construction process with python execution for answer generation seems to guarantee that. What's the justification behind limiting it to these types of questions? How are you prompting the LLM to generate questions?
>
> Thank you for raising the point. Our focus on Python-answerable questions stems from our goal of enabling quantitative analysis and visualization directly from tabular data. Python offers a flexible environment for data manipulation, allowing a wide range of meaningful questions to be automatically answered through code execution. Furthermore, the question types we adopt are designed to cover a comprehensive space of quantitative reasoning tasks, consistent with prior work such as MMQA [2] and TQA-Bench [3], while extending them to more practical scenarios involving visualization and multi-turn analysis. This design choice allows the benchmark to remain grounded in tasks that are both answerable in an automated, reproducible manner and representative of real-world analytical workflows.
>
> To generate questions, we provide the LLM with the table information, the relevant external knowledge, and the name and description of the question type. We will point to Prompt 1 in Appendix E more clearly in the revision so that readers can understand the question generation stage.
>
> [2] Jian Wu, Linyi Yang, Dongyuan Li, Yuliang Ji, Manabu Okumura, and Yue Zhang. MMQA: Evalu-
> ating LLMs with multi-table multi-hop complex questions. In ICLR, 2025
>
> [3] Zipeng Qiu, You Peng, Guangxin He, Binhang Yuan and Chen Wang. TQA-Bench: Evaluating LLMs for Multi-Table Question Answering with Scalable Context and Symbolic Extension. 2024, https://arxiv.org/abs/2411.19504

---

> ### Author Response · Authors · 2025-11-25
> **Author Response to Reviewer UM5Z (Part 3)**
>
> > I appreciate the authors acknowledged the potential subjectivity of insights and their attempt to address it. I'm curious if the authors have conducted analysis on annotator agreement for the generated insights
>
> Thank you for raising this important point regarding annotator agreement. We indeed conducted an analysis to examine how well the proposed insight-level scores align with human judgments.
>
> Specifically, we sampled 50 pairs of generated insights (Insight A and Insight B) for each ground-truth insight and asked three independent annotators to assess which of the two was closer to the ground truth. Annotators selected one of five relative comparative options: A+ (A is definitely better), A (A is slightly better), N (similar), B (B is slightly better), or B+ (B is definitely better). Each option was mapped to a normalized score in {-2, −1, 0, 1, 2}. We averaged the three annotators’ scores to obtain the final human-judgment score for each pair.
>
> In parallel, we computed a normalized score from the insight-level metric based on the difference in metric scores (score for B minus score for A). We then evaluated the agreement between human judgments and the metric-derived scores using both Pearson and Spearman correlations. The resulting correlations were 0.669 and 0.663, respectively. These results indicate that the proposed insight-level metric exhibits a strong alignment with human perception, supporting its validity as an automatic proxy for human preference.
>
> We will include the evaluation in Section 4.1.
>
> > "After executing the code from all four models, we measured the answer consensus to filter out questions that yielded unanimous agreement across all four LLMs. Such instances were deemed to indicate a low level of analytical complexity, making them unsuitable for a benchmark to challenge state-of-the-art models." how do you measure answer consensus?
>
> As for the text-based answer, exact matching is used to check the consensus. When it comes to the visualization, the annotation GUI (Figure 7) is used to assess whether all the generated visualizations are similar or not. We will clarify the measurement of consensus in the revised version in Appendix B.3.
>
> > what is the justification for the claim that 'unanimous agreement' indicates 'low level of analytical complexity' -- seems like this is just to intentionally construct an adversarial set that is hard for current systems, not so much representative of true distribution/human interests?
>
> Thank you for raising this important concern. We would like to clarify that our decision to exclude “unanimous agreement” cases is not intended to construct an adversarial benchmark.
>
> Since these questions can be resolved reliably across models with a small number of reasoning steps (e.g. lookup with single column filtering, aggregation of single column), we consider them to have low analytical complexity that are not observed in the real-world deployment. Therefore, the questions are less informative for evaluating multi-step reasoning over heterogeneous large multi-tabular data.
>
> Importantly, among 1840 generated questions after the question scoring stage, only 114 questions (~6%) correspond to the “unanimous agreement” cases, and the remaining 94% questions are passed onto the human verification process, which does not significantly distort the true distribution of generated questions. Thus, the resulting dataset is representative of non-trivial analytical scenarios encountered in real-world data analysis, rather than an intentionally adversarial or inflated difficulty set.
>
> > if the executed code has no agreement, how do you decide which answer to keep?
>
> Annotator checked each generated code by LLMs with the use of GUI (Figure 7), and decided to modify (not keep) the code so that it produces plausible answers.
>
> > Human verification stage -- "During this stage, annotators also filtered out questions for qualitative reasons, such as not being insightful, being too ambiguous to permit a definitive answer, or requiring external knowledge that was unavailable." => what is considered 'insightful'?
>
> Thank you for the important clarification question to reproduce the dataset construction. In our human verification stage, a question is considered not insightful when it does not lead to an analytically meaningful or practically useful interpretation of the dataset. For example, some generated questions produce results that are technically computable but lack substantive analytical value in the context of the dataset. A typical case arises in geospatial datasets: questions such as “What is the average latitude and longitude under certain conditions?” often result in a point located in an geographically meaningless location (e.g., in the ocean), providing no useful insight for downstream analysis. Questions of this nature were excluded. We will elucidate the point in the revised manuscript in Appendix B.3.

---

### Meta-Review · Area_Chair_WEQr · 2026-01-07

**Summary:**

The authors proposed, OpenDataBench, which introduces a benchmark built from governmental open data to evaluate (i) multifaceted Table QA over large / multi-table data (including text or visualization outputs) and (ii) Table Insight generation for exploratory analysis, alongside proposed Answer/Insight “agents” and baseline evaluations showing current LLMs struggle on these settings. Reviewers generally agreed the benchmark targets an important, realistic setting, but have the concern on the proposed agent approach and the limitation of the evaluation and novelty. The authors tackle an important problem for table understanding with large models, but it may need more insights through benchmarking to stand out.

**Reviewer Concerns:**

Addressed:

UM5Z: the concerns on implementation details are mostly addressed (insight extraction, baselines, stopping criteria ... )
32SQ: 10-row baselines details, SQL integration
n5Wt: correlation

Outstanding:
UM5Z: key soundness details
n5Wt: insights from the benchmark
SPHH: novelty

**Reviewer Scores:**

All reviewers may raise the score by 1

---

### Decision · Program_Chairs · 2026-01-26

Reject